# On Sample Complexity of Offline Reinforcement Learning with Deep ReLU Networks in Besov Spaces

**Thanh Nguyen-Tang**                                    *thnguyentang@gmail.com*
*Department of Computer Science, Johns Hopkins University*

**Sunil Gupta & Hung Tran-The & Svetha Venkatesh**
*Applied AI Institute, Deakin University*

Reviewed on OpenReview: *https://openreview.net/forum?id=LdEmOumNcv*

## Abstract

Offline reinforcement learning (RL) leverages previously collected data for policy optimization without any further active exploration. Despite the recent interest in this problem, its theoretical results in neural network function approximation settings remain elusive. In this paper, we study the statistical theory of offline RL with deep ReLU network function approximation. In particular, we establish the sample complexity of $n = \tilde{\mathcal{O}}(H^{4+4\frac{d}{\alpha}}\kappa_\mu^{1+\frac{d}{\alpha}}\epsilon^{-2-2\frac{d}{\alpha}})$ for offline RL with deep ReLU networks, where $\kappa_\mu$ is a measure of distributional shift, $H = (1-\gamma)^{-1}$ is the effective horizon length, $d$ is the dimension of the state-action space, $\alpha$ is a (possibly fractional) smoothness parameter of the underlying Markov decision process (MDP), and $\epsilon$ is a user-specified error. Notably, our sample complexity holds under two novel considerations: the Besov dynamic closure and the correlated structure. While the Besov dynamic closure subsumes the dynamic conditions for offline RL in the prior works, the correlated structure renders the prior works of offline RL with general/neural network function approximation improper or inefficient in long (effective) horizon problems. To the best of our knowledge, this is the first theoretical characterization of the sample complexity of offline RL with deep neural network function approximation under the general Besov regularity condition that goes beyond the linearity regime in the traditional Reproducing Hilbert kernel spaces and Neural Tangent Kernels.

## 1 Introduction

Offline reinforcement learning (Lange et al., 2012; Levine et al., 2020) is a practical paradigm of reinforcement learning (RL) where logged experiences are abundant but a new interaction with the environment is limited or even prohibited. The fundamental offline RL problems concern with how well previous experiences could be used to evaluate a new target policy, known as off-policy evaluation (OPE) problem, or to learn the optimal policy, known as off-policy learning (OPL) problem. We study these offline RL problems with infinitely large state spaces, where the agent must rely on function approximation such as deep neural networks to generalize across states from an offline dataset. Such problems form the core of modern RL in practical settings (Levine et al., 2020; Kumar et al., 2020; Singh et al., 2020; Zhang et al., 2022).

Prior sample-efficient results in offline RL mostly focus on tabular environments with small finite state spaces (Yin & Wang, 2020; Yin et al., 2021; Yin & Wang, 2021a), but as these methods scale with the number of states, they are infeasible for the settings with infinitely large state spaces. While this tabular setting has been extended to large state spaces via linear environments (Duan & Wang, 2020; Jin et al., 2020b; Xiong et al., 2022; Yin et al., 2022; Nguyen-Tang et al., 2022b), the linearity assumption often does not hold for many RL problems in practice. Theoretical guarantees for offline RL with general and deep neural network function approximations have also been derived, but these results are either inadequate or relatively disconnected from the regularity structure of the underlying MDP. In particular, while the finite-sample results for offline

RL with general function approximation (Munos & Szepesvári, 2008; Le et al., 2019) depend on an inherent Bellman error which could be uncontrollable in practice, the other analysis in the neural network function approximation in Yang et al. (2019) relies on a data splitting technique to deal with the correlated structures arisen in value regression for offline RL and use a relatively strong dynamic assumption. Recent works have studied offline RL with function approximations in Reproducing Hilbert kernel spaces (RHKS) (Jin et al., 2020b) and Neural Tangent Kernels (NTK) (Nguyen-Tang et al., 2022a). However, these function classes also have (approximately) linear structures (in terms of the underlying features) which make their analysis similar to the linear case. Moreover, the smoothness assumption imposed by the RKHS is often strong for several practical cases while the NTK analysis requires a extremely wide neural net (the network width scales with $n^{10}$ for the NTK case in Nguyen-Tang et al. (2022a) versus only $n^{2/5}$ (Proposition 5.1) in the current work). Recent works (Xie et al., 2021a; Zhan et al., 2022; Chen & Jiang, 2022; Uehara & Sun, 2021) have considered offline RL with general function approximation with weaker data coverage assumption. However, they assumed the function class is finite and did not consider the (Besov) regularity of the underlying MDP. Thus, to our knowledge, no prior work has dedicated to study a comprehensive and adequate analysis of the statistical efficiency for offline RL with neural network function approximation in Besov spaces. Thus, it is natural to ask:

*Is offline RL sample-efficient with deep ReLU network function approximation beyond the (approximate-) linear regime imposed by RKHS and NTK?*

**Our contributions.** In this paper, we provide a statistical theory of both OPE and OPL with neural network function approximation in a broad generality that is beyond the (approximate-) linear regime imposed by RKHS and NTK. In particular, our contributions, which are summarized in Table 1 and will be discussed in details in Section 5, are:

- First, we achieve a generality for the guarantees of offline RL with neural network function approximation via two novel considerations: (i) We introduce a new structural condition namely *Besov dynamic closure* which subsumes the existing dynamic conditions for offline RL with neural network function approximation and even includes MDPs that need not be continuous, differentiable or spatially homogeneous in smoothness; (ii) We take into account the correlated structure of the value estimate produced by a regression-based algorithm from the offline data. This correlated structure plays a central role in the statistical efficiency of an offline algorithm; yet the prior works improperly ignore this structure (Le et al., 2019) or avoid it using an data splitting approach (Yang et al., 2019).

- Second, we prove that an offline RL algorithm based on fitted-Q iteration (FQI) can achieve the sample complexity of $n = \tilde{\mathcal{O}}(H^{4+4\frac{d}{\alpha}}\kappa_\mu^{1+\frac{d}{\alpha}}\epsilon^{-2-2\frac{d}{\alpha}})$ where $\kappa$ measures the distributional shift in the offline data, $H = (1-\gamma)^{-1}$ is the effective horizon length, $d$ is the input dimension, $\alpha$ is a smoothness parameter of the underlying MDP, and $\epsilon$ is a user-specified error. Notably, our guarantee holds under our two novel considerations above that generalize the condition in Yang et al. (2019) and do not require data splitting in Yang et al. (2019). Moreover, our analysis also corrects the technical mistake in Le et al. (2019) that ignores the correlated structure of offline value estimate.

**Problem scope.** We emphasize that the present work focuses on statistical theory of offline RL with neural network function approximation in Besov spaces where we analyze a relatively standard algorithm, FQI. Regarding the empirical effectiveness of FQI with neural network function approximation for offline RL, we refer the readers to the empirical study in Voloshin et al. (2019). Finally, this work is an extension of our workshop paper (Nguyen-Tang et al., 2021).

**Notations.** Denote $\|f\|_{p,\mu} := \left(\int_{\mathcal{X}} |f|^p d\mu\right)^{1/p}$, and for simplicity, we write $\|\cdot\|_\mu$ for $\|\cdot\|_{p,\mu}$ when $p = 2$ and write $\|\cdot\|_p$ for $\|\cdot\|_{p,\mu}$ if $\mu$ is the Lebesgue measure. Let $L^p(\mathcal{X})$ be the space of measurable functions for which the $p$-th power of the absolute value is Lebesgue integrable, i.e. $L^p(\mathcal{X}) = \{f : \mathcal{X} \to \mathbb{R} | \|f\|_p < \infty\}$. Denote by $\|\cdot\|_0$ the 0-norm, i.e., the number of non-zero elements, and $a \vee b = \max\{a, b\}$. For any two

Table 1: Comparison among existing representative works of FQI estimators for offline RL with function approximation under a *uniform* data coverage assumption. Here $S$ and $A$ are the cardinalities of the state and action space when they are finite, $\kappa$ is a measure of distribution shift (which can be defined slightly different in different works), $\epsilon$ is the user-specified precision, $d$ is the dimension of the input space, $\alpha$ is the smoothness parameter of the underlying MDP, and $H := (1-\gamma)^{-1}$ is the effective horizon length.

| Work | Function | Regularity | Tasks | Sample complexity | Remark |
|------|----------|-----------|-------|-------------------|--------|
| Yin & Wang (2020) | Tabular | Tabular | OPE | $\tilde{\mathcal{O}}\left(\kappa \cdot H^4 \cdot \epsilon^{-2} \cdot (SA)^2\right)$ | - |
| Duan & Wang (2020) | Linear | Linear | OPE | $\tilde{\mathcal{O}}\left(\kappa \cdot H^4 \cdot \epsilon^{-2} \cdot d\right)$ | - |
| Le et al. (2019) | General | General | OPE/OPL | N/A | improper analysis |
| Yang et al. (2019) | ReLU nets | Hölder | OPL | $\tilde{\mathcal{O}}\left(\kappa^{2+\frac{d}{\alpha}} \cdot H^{5+2\frac{d}{\alpha}} \cdot \epsilon^{-2-\frac{d}{\alpha}} \cdot \log(H^2/\epsilon)\right)$ | data splitting |
| This work | ReLU nets | Besov | OPE/OPL | $\tilde{\mathcal{O}}\left(\kappa^{1+\frac{d}{\alpha}} \cdot H^{4+4\frac{d}{\alpha}} \cdot \epsilon^{-2-2\frac{d}{\alpha}}\right)$ | data reuse |

real-valued functions $f$ and $g$, we write $f(\cdot) \lesssim g(\cdot)$ if there is an absolute constant $c$ independent of the function parameters $(\cdot)$ such that $f(\cdot) \leq c \cdot g(\cdot)$. We write $f(\cdot) \asymp g(\cdot)$ if $f(\cdot) \lesssim g(\cdot)$ and $g(\cdot) \lesssim f(\cdot)$. We write $f(\cdot) \simeq g(\cdot)$ if there exists an absolute constant $c$ such that $f(\cdot) = c \cdot g(\cdot)$. We denote $H := (1-\gamma)^{-1}$ which is the effective horizon length in the discounted MDP and is equivalent to the horizon (episode) length in finite-horizon MDPs.

## 2 Related Work

**Offline RL with tabular representation.** The majority of the theoretical results for offline RL focus on tabular MDP where the state space is finite and an importance sampling -related approach is possible (Precup et al., 2000; Dudík et al., 2011; Jiang & Li, 2015; Thomas & Brunskill, 2016; Farajtabar et al., 2018; Kallus & Uehara, 2019). The main drawback of the importance sampling-based approach is that it suffers high variance in long horizon problems. The high variance problem can be mitigated by direct methods where we employ models to estimate the value functions or the transition kernels. We focus on direct methods in this work. For tabular MDPs with some uniform data-visitation measure $d_m > 0$, a near-optimal sample complexity bound of $\mathcal{O}(H^3 d_m/\epsilon^2)$ and $\mathcal{O}(H^2 d_m/\epsilon^2)$ were obtained for time-inhomogeneous tabular MDP (Yin et al., 2021) and for time-homogeneous tabular MDP (Yin & Wang, 2021b; Ren et al., 2021), respectively. With the single-concentrability assumption, a tight bound of $\mathcal{O}(H^3 SC^*/\epsilon^2)$ was achieved (Xie et al., 2021b; Rashidinejad et al., 2021), where $H \approx 1/(1-\gamma)$ is the episode length. Yin & Wang (2021c) introduced intrinsic offline bound that further incorporates instance-dependent quantities. Shi et al. (2022) obtained the minimax rate with model-free methods. Wang et al. (2022) derived gap-dependent bounds for offline RL in tabular MDPs.

**Offline RL with linear function approximation.** Offline RL with function approximation often follow two algorithmic approaches: Fitted Q-iteration (FQI) (Bertsekas & Tsitsiklis, 1995; Jong & Stone, 2007; Lagoudakis & Parr, 2003; Grünewälder et al., 2012; Munos, 2003; Munos & Szepesvári, 2008; Antos et al., 2008; Tosatto et al., 2017; Le et al., 2019), and pessimism principle (Buckman et al., 2020), where the former requires a uniform data coverage and the latter only needs a sufficient coverage over the target policy. Duan & Wang (2020) studied fitted-Q iteration algorithm in linear MDPs. Wang et al. (2020) highlighted the necessity of strong structural assumptions (e.g., on low distributional shift or strong dynamic condition beyond realizability) for sample-efficient offline RL with linear function approximation suggesting that only realizability and strong uniform data coverage are not sufficient for sample-efficient offline RL. Jin et al. (2020b) brought pessimism principle into offline linear MDPs. Nguyen-Tang et al. (2022b) derived a minimax rate of $1/\sqrt{n}$ for offline linear MDPs under a partial data coverage assumption and obtained the instance-dependent rate of $1/n$ when the gap information is available. Xiong et al. (2022); Yin et al. (2022) used variance reduction and data splitting to tighten the bound of Jin et al. (2020b). Xie et al. (2021a) proposed

Bellman-consistent condition with general function approximation which improves the bound of Jin et al. (2020b) by a factor of $\sqrt{d}$ when realized to finite action space and linear MDPs. Chen et al. (2021) studied sample complexity of FQI in linear MDPs and derive a lower bound for this setting.

**Offline RL with non-linear function approximation.** Beyond linearity, some works study offline RL in general or nonparametric function approximation, either with FQI estimators (Munos & Szepesvári, 2008; Le et al., 2019; Duan et al., 2021a;b; Hu et al., 2021), pessimistic estimators (Uehara & Sun, 2021; Nguyen-Tang et al., 2022a; Jin et al., 2020b), or minimax estimators (Uehara et al., 2021), where Uehara et al. (2021) also realized their minimax estimators to the neural network function approximation, Nguyen-Tang et al. (2022a) considered offline contextual bandits with Neural Tangent Kernels (NTK), and Jin et al. (2020b) considered the pessimistic value iteration algorithm with Reproducing Kernel Hilbert Space (RKHS) in their extended version. Our work is different from these aforementioned works in that we analyze the fundamental FQI estimators with neural network function approximation under the Besov regularity condition that is much more general than RKHS and NTK. We also further emphasize that even that RKHS and NTK spaces are non-linear function approximation, the functions in those spaces are linear in terms of an underlying feature space, making the analysis for these spaces akin to the case of linear function approximation. Yang et al. (2019) also considered deep neural network approximation. In particular, Yang et al. (2019) focused on analyzing deep Q-learning using a disjoint fold of offline data for each iteration. Such approach is considerably sample-inefficient for offline RL with long (effective) horizon. In addition, they rely on a relatively restricted smoothness assumption of the underlying MDPs that hinders their results from being widely applicable in more general settings. Recently, other works (Xie et al., 2021a; Zhan et al., 2022; Chen & Jiang, 2022; Uehara & Sun, 2021) considered offline RL with general function approximation and imposed weaker data coverage assumption by using pessimistic algorithms. Their algorithms are more involved than FQI but did not study the effect of the regularity of the underlying MDP on the sample complexity of offline RL. They also assume that the function class is finite which is not applicable to neural network function approximation. Since the first version of our paper appeared online, there have been several other works establishing sample complexity of reinforcement learning in Besov spaces for various problem settings, including $\epsilon$-greedy exploration for online setting with Markovian data (Liu et al., 2022) and off-policy evaluation on low-dimensional manifolds (Ji et al., 2022).

## 3 Preliminaries

We consider a discounted Markov decision process (MDP) with possibly infinitely large state space $\mathcal{S}$, continuous action space $\mathcal{A}$, initial state distribution $\rho \in \mathcal{P}(\mathcal{S})$, transition operator $P : \mathcal{S} \times \mathcal{A} \to \mathcal{P}(\mathcal{S})$, reward distribution $R : \mathcal{S} \times \mathcal{A} \to \mathcal{P}([0, 1])$, and a discount factor $\gamma \in [0, 1)$. For notational simplicity, we assume that $\mathcal{X} := \mathcal{S} \times \mathcal{A} \subseteq [0, 1]^d$.

A policy $\pi : \mathcal{S} \to \mathcal{P}(\mathcal{A})$ induces a distribution over the action space conditioned on states. The $Q$-value function for policy $\pi$ at state-action pair $(s, a)$, denoted by $Q^\pi(s, a) \in [0, 1]$, is the expected discounted total reward the policy collects if it initially starts in the state-action pair, i.e.,

$$Q^\pi(s, a) := \mathbb{E}_\pi \left[ \sum_{t=0}^\infty \gamma^t r_t | s_0 = s, a_0 = a \right],$$

where $r_t \sim R(s_t, a_t), a_t \sim \pi(\cdot|s_t)$, and $s_t \sim P(\cdot|s_{t-1}, a_{t-1})$. The value of policy $\pi$ is $V^\pi = \mathbb{E}_{s \sim \rho, a \sim \pi(\cdot|s)} [Q^\pi(s, a)]$, and the optimal value is $V^* = \max_\pi V^\pi$ where the maximization is taken over all stationary policies. Alternatively, the optimal value $V^*$ can be obtained via the optimal $Q$-function $Q^* = \max_\pi Q^\pi$ as $V^* = \mathbb{E}_{s \sim \rho} [\max_a Q^*(s, a)]$. Denote by $T^\pi$ and $T^*$ the Bellman operator and the optimality Bellman operator, respectively, i.e., for any $f : \mathcal{S} \times \mathcal{A} \to \mathbb{R}$

$$[T^\pi f](s, a) = \mathbb{E}_{r \sim R(s, a)}[r] + \gamma \mathbb{E}_{s' \sim P(\cdot|s, a), a' \sim \pi(\cdot|s')} [f(s', a')]$$

$$[T^* f](s, a) = \mathbb{E}_{r \sim R(s, a)}[r] + \gamma \mathbb{E}_{s' \sim P(\cdot|s, a)} \left[ \max_{a'} f(s', a') \right],$$

we have $T^\pi Q^\pi = Q^\pi$ and $T^* Q^* = Q^*$.

**Offline regime.** We consider the offline RL setting where a learner cannot explore the environment but has access to a fixed logged data $\mathcal{D} = \{(s_i, a_i, s_i', r_i)\}_{i=1}^n$ collected a priori by certain behaviour policy $\eta$. For simplicity, we assume that $\{s_i\}_{i=1}^n$ are independent and $\eta$ is stationary. Equivalently, $\{(s_i, a_i)\}_{i=1}^n$ are i.i.d. samples from the normalized discounted stationary distribution over state-actions with respect to $\eta$, i.e., $(s_i, a_i) \overset{i.i.d.}{\sim} \mu(\cdot, \cdot) := (1 - \gamma) \sum_{t=0}^\infty \gamma^t \mathbb{P}(s_t = \cdot, a_t = \cdot | \rho, \eta)$ where $s_i' \sim P(\cdot | s_i, a_i)$ and $a_i \sim \eta(\cdot | s_i)$. This assumption is relatively standard in the offline RL setting (Munos & Szepesvári, 2008; Chen & Jiang, 2019a; Yang et al., 2019). The goals of Off-Policy Evaluation (OPE) and Off-Policy Learning (OPL) are to estimate $V^\pi$ and $V^*$, respectively from $\mathcal{D}$. The performance of OPE and OPL estimates are measured via sub-optimality gaps defined as follows.

**For OPE Task.** Given a fixed target policy $\pi$, for any value estimate $\hat{V}$ computed from the offline data $\mathcal{D}$, the sub-optimality of OPE is defined as

$$\text{SubOpt}(\hat{V}; \pi) = |V^\pi - \hat{V}|.$$

**For OPL Task.** For any estimate $\hat{\pi}$ of the optimal policy $\pi^*$ that is learned from the offline data $\mathcal{D}$, we define the sup-optimality of OPL as

$$\text{SubOpt}(\hat{\pi}) := \mathbb{E}_{s \sim \rho} \left[ V^*(s) - V^{\hat{\pi}}(s) \right].$$

## 3.1 Deep ReLU Networks as Function Approximation

In practice, the state space is often very large and complex, and thus function approximation is required to ensure generalization across different states. Deep neural networks with the ReLU activation offer a rich class of parameterized functions with differentiable parameters. Deep ReLU networks are state-of-the-art in many applications, e.g., Krizhevsky et al. (2012); Mnih et al. (2015), including offline RL with deep ReLU networks that can yield superior empirical performance (Voloshin et al., 2019). In this section, we describe the architecture of deep ReLU networks and the associated function space which we use throughout this paper. Specifically, a $L$-height, $m$-width ReLU network on $\mathbb{R}^d$ takes the form of

$$f_\theta^{L,m}(x) = W^{(L)} \sigma \left( W^{(L-1)} \sigma \left( \ldots \sigma \left( W^{(1)} \sigma(x) + b^{(1)} \right) \ldots \right) + b^{(L-1)} \right) + b^{(L)},$$

where $W^{(L)} \in \mathbb{R}^{1 \times m}, b^{(L)} \in \mathbb{R}, W^{(1)} \in \mathbb{R}^{m \times d}, b^{(1)} \in \mathbb{R}^m, W^{(l)} \in \mathbb{R}^{m \times m}, b^{(l)} \in \mathbb{R}^m, \forall 1 < l < L, \theta = \{W^{(l)}, b^{(l)}\}_{1 \leq l \leq L}$, and $\sigma(x) = \max\{x, 0\}$ is the (element-wise) ReLU activation. We define $\Phi(L, m, S, B)$ as the space of $L$-height, $m$-width ReLU functions $f_\theta^{L,m}(x)$ with sparsity constraint $S$, and norm constraint $B$, i.e., $\sum_{l=1}^L (\|W^{(l)}\|_0 + \|b^{(l)}\|_0) \leq S, \max_{1 \leq l \leq L} \|W^{(l)}\|_\infty \vee \|b^{(l)}\|_\infty \leq B$. Finally, for some $L, m \in \mathbb{N}$ and $S, B \in (0, \infty)$, we define the unit ball of ReLU network function space $\mathcal{F}_{NN}$ as

$$\mathcal{F}_{\text{NN}}(L, m, S, B) := \left\{ f \in \Phi(L, m, S, B) : \|f\|_\infty \leq 1 \right\}.$$

In nonparametric regressions, Suzuki (2018) showed that deep ReLU networks outperform any non-adaptive linear estimator due to their higher adaptivity to spatial inhomogeneity.

## 3.2 Besov spaces

Our new dynamic condition relies on the regularity of Besov spaces. There are several ways to characterize the smoothness in Besov spaces. Here, we pursue a characterization via multivariate moduli of smoothness as it is more intuitive, following Giné & Nickl (2016).

**Definition 3.1** (*Multivariate moduli of smoothness*). *For any $t > 0$ and $r \in \mathbb{N}$, the $r$-th multivariate modulus of smoothness of any function $f \in L^p(\mathcal{X}), p \in [1, \infty]$ is defined as*

$$\omega_r^{t,p}(f) := \sup_{0 \leq \|h\| \leq t} \|\Delta_h^r(f)\|_p,$$

where $\Delta_h^r(f)$ is the $r$-th order translation-difference operator defined as

$$\Delta_h^r(f)(\cdot) := \sum_{k=0}^{r} \binom{r}{k} (-1)^{r-k} f(\cdot + k \cdot h).$$

*Remark* 3.1. The quantity $\Delta_h^r(f)$ captures the local oscillation of $f$ which is not necessarily differentiable. In the case the $r$-th order weak derivative $D^r f$ exists and is locally integrable, we have

$$\lim_{h \to 0} \frac{\Delta_h^r(f)(x)}{\|h\|^r} = D^r f(x).$$

It also follows from Minkowski's inequality that

$$\frac{\omega_r^{t,p}(f)}{t^r} \le \|D^r f\|_p \text{ and } \frac{\omega_{r+r'}^{t,p}(f)}{t^r} \le \omega_{r'}^{t,p}(D^r f).$$

**Definition 3.2** (Besov space $B_{p,q}^\alpha(\mathcal{X})$). *For $1 \le p, q \le \infty$ and $\alpha > 0$, we define the norm $\|\cdot\|_{B_{p,q}^\alpha}$ of the Besov space $B_{p,q}^\alpha(\mathcal{X})$ as $\|f\|_{B_{p,q}^\alpha} := \|f\|_p + |f|_{B_{p,q}^\alpha}$ where*

$$|f|_{B_{p,q}^\alpha} := \begin{cases} \left( \int_{\mathbb{R}_+} \left( \frac{\omega_{\lfloor \alpha \rfloor + 1}^{t,p}(f)}{t^\alpha} \right)^q \frac{dt}{t} \right)^{1/q}, & 1 \le q < \infty, \\ \sup_{t > 0} \frac{\omega_{\lfloor \alpha \rfloor + 1}^{t,p}(f)}{t^\alpha}, & q = \infty, \end{cases}$$

*is the Besov seminorm. Then, $B_{p,q}^\alpha := \{f \in L^p(\mathcal{X}) : \|f\|_{B_{p,q}^\alpha} < \infty\}$.*

Intuitively, the Besov seminorm $|f|_{B_{p,q}^\alpha}$ roughly describes the $L^q$-norm of the $l^p$-norm of the $\alpha$-order smoothness of $f$. Besov spaces are considerably general that subsume Hölder spaces and Sobolev spaces as well as functions with spatially inhomogeneous smoothness (Triebel, 1983; Sawano, 2018; Suzuki, 2018; Cohen, 2009; Nickl & Pötscher, 2007). In particular, the Besov space $B_{p,q}^\alpha$ reduces into the Hölder space $C^\alpha$ when $p = q = \infty$ and $\alpha$ is positive and non-integer while it reduces into the Sobolev space $W_2^\alpha$ when $p = q = 2$ and $\alpha$ is a positive integer. We further consider the unit ball of $B_{p,q}^\alpha(\mathcal{X})$ as $\bar{B}_{p,q}^\alpha(\mathcal{X}) := \{g \in B_{p,q}^\alpha : \|g\|_{B_{p,q}^\alpha} \le 1 \text{ and } \|g\|_\infty \le 1\}$. When the context is clear, we drop $\mathcal{X}$ from $\bar{B}_{p,q}^\alpha(\mathcal{X})$.

## 4 Fitted Q-Iteration for Offline Reinforcement Learning

---

**Algorithm 1** Fitted Q-Iteration with Neural Network Function Approximation

---

1: **Input:** Offline data $\mathcal{D} = \{(s_i, a_i, s_i', r_i)\}_{i=1}^n$, number of iterations $K$, function family $\mathcal{F}_{\text{NN}}$, target policy $\pi$ (for OPE Task only)
2: Initialize $Q_0 \in \mathcal{F}_{\text{NN}}$
3: **for** $k = 1, \ldots, K$ **do**
4:  Compute the estimated state-action value $Q_k$ as

$$\begin{cases} Q_k \leftarrow \arg\min_{f \in \mathcal{F}_{\text{NN}}} \frac{1}{n} \sum_{i=1}^n \left( f(s_i, a_i) - r_i - \gamma \mathbb{E}_{a' \sim \pi(\cdot|s_i')} \left[ Q_{k-1}(s_i', a) \right] \right)^2 & \text{if OPE Task} \\ Q_k \leftarrow \arg\min_{f \in \mathcal{F}_{\text{NN}}} \frac{1}{n} \sum_{i=1}^n \left( f(s_i, a_i) - r_i - \gamma \max_{a \in \mathcal{A}} Q_{k-1}(s_i', a) \right)^2 & \text{if OPL Task} \end{cases}$$

5: **end for**
6: **Output:** Return the following estimates

$$\begin{cases} V_K \leftarrow \|Q_K\|_{\rho^\pi} := \sqrt{\mathbb{E}_{\rho(s)\pi(a|s)} \left[ Q_K(s,a)^2 \right]} & \text{if OPE Task} \\ \pi_K(\cdot|s) \leftarrow \arg\max_a Q_K(a|s) & \text{if OPL Task} \end{cases}$$

---

In this work, we study a variant of fitted Q-iteration (FQI) algorithm for offline RL, presented in Algorithm 1. This algorithm is appealingly simple as it iteratively constructs $Q$-estimate from the offline data and the previous $Q$-estimate, as in Algorithm 1. This FQI-style algorithm has been largely studied for offline RL, such as Munos & Szepesvári (2008); Chen & Jiang (2019a); Duan et al. (2021a) to name a few; yet there has been no work studying this algorithm in offline RL with neural network function approximation except Yang et al. (2019). However, Yang et al. (2019) use data splitting and rely on a more limited dynamic condition than ours. Thus, the notable difference in Algorithm 1 is the use of neural network to approximate $Q$-functions and we estimate each $Q_k$ using the entire offline data set, instead of splitting the data into disjoint sets as in Yang et al. (2019). In particular, Yang et al. (2019) split the offline data into $K$ disjoint sets, resulting in the sample complexity linearly scaled with $K$, which is highly inefficient in long (effective) horizon problems where the effective horizon length $H = 1/(1 - \gamma)$ is large.

As we do not split the data into disjoint sets, a correlated structure is induced. Specifically, at each iteration $k$ in Algorithm 1, $Q_{k-1}$ also depends on $(s_i, a_i)$ which makes $\mathbb{E}\left[r_i + \gamma \max_a Q_{k-1}(s_i', a)\right] \neq [T^* Q_{k-1}](s_i, a_i)$ in OPL Task (and $\mathbb{E}\left[r_i + \gamma \mathbb{E}_{a \sim \pi(\cdot|s_i')}[Q_{k-1}(s_i', a)]\right] \neq [T^\pi Q_{k-1}](s_i, a_i)$ in OPE Task, respectively). This correlated structure hinders a direct use of the standard concentration inequalities such as Bernstein's inequality that require a sequence of random variables to adapt to certain filtration. We overcome this technical difficulty using uniform convergence argument.

In our analysis, we assume access to the minimizer of the optimization in Algorithm 1. In practice, we can use (stochastic) gradient descent to effectively solve this optimization with $L_0$ regularization (Louizos et al., 2017). If the $L_0$ constraint is relaxed in practice, (stochastic) gradient descent is guaranteed to converge to a global minimum under certain structural assumptions (Du et al., 2019a;b; Allen-Zhu et al., 2019; Nguyen, 2021).

## 5 Main Result

To obtain a non-trivial guarantee, certain assumptions on the distribution shift and the MDP regularity are necessary. We introduce the assumptions about the data generation in Assumption 5.1 and the regularity of the underlying MDP 5.2.

**Assumption 5.1** (*Uniform concentrability coefficient (Munos & Szepesvári, 2008)*)**.** *$\exists \kappa_\mu < \infty$ such that* $\left\|\frac{d\nu}{d\mu}\right\|_\infty \leq \kappa_\mu$ *for any admissible distribution $\nu$.* [1]

The finite $\kappa_\mu$ in Assumption 5.1 asserts that the sampling distribution $\mu$ is not too far away from any admissible distribution, which holds for a reasonably large class of MDPs, e.g., for any finite MDP, any MDP with bounded transition kernel density, and equivalently any MDP whose top-Lyapunov exponent is negative. We present a simple (though stronger than necessary) example for which Assumption 5.1 holds.

**Example 5.1.** *If there exist absolute constants $c_1, c_2 > 0$ such that for any $s, s' \in \mathcal{S}$, there exists an action $a \in \mathcal{A}$ such that $P(s'|s, a) \geq 1/c_1$ and $\eta(a|s) \geq 1/c_2, \forall s, a$, then we can choose $\kappa_\mu = c_1 c_2$.*

Chen & Jiang (2019a) further provided natural problems with rich observations generated from hidden states that has a low concentrability coefficient. These suggest that low concentrability coefficients can be found in fairly many interesting problems in practice.

We now state the assumption about the regularity of the underlying MDP.

**Assumption 5.2** (*Besov dynamic closure*)**.** *Consider some fixed $p, q \in [1, \infty]$ and $\alpha > \frac{d}{\min\{p, 2\}}$.*

- *For OPE Task: For a target policy $\pi$, and for some $(L, m, S, B) \in \mathbb{N} \times \mathbb{N} \times \mathbb{N} \times \mathbb{R}_+$ (which will be specified later) we assume that: $\forall f \in \mathcal{F}_{\mathrm{NN}}(L, m, S, B) \implies T^\pi f \in \bar{B}_{p,q}^\alpha$.*

- *For OPL Task: For some $(L, m, S, B) \in \mathbb{N} \times \mathbb{N} \times \mathbb{N} \times \mathbb{R}_+$ (which will be specified later) we assume that: $\forall f \in \mathcal{F}_{\mathrm{NN}}(L, m, S, B) \implies T^* f \in \bar{B}_{p,q}^\alpha$.*

---

[1]$\nu$ is said to be admissible if there exist $t \geq 0$ and policy $\bar{\pi}$ such that $\nu(s, a) = \mathbb{P}(s_t = s, a_t = a|s_1 \sim \rho, \bar{\pi}), \forall s, a$.

Assumption 5.2 signifies that for OPL task (for OPE task with target policy $\pi$, respectively) the Bellman operator $T^*$ ($T^\pi$, respectively) applied on any ReLU network function in $\mathcal{F}_{\mathrm{NN}}(L, m, S, B)$ results in a new function that sits in $\bar{B}^\alpha_{p,q}(\mathcal{X})$. The smoothness constraint $\alpha > \frac{d}{\min\{p,2\}}$ is necessary to guarantee the compactness and the finite (local) Rademacher complexity of the Besov space, and $\alpha - d/p$ is called the *differential dimension* of the Besov space. Note that when $p < 2$ (thus the condition above becomes $\alpha > d/p$), a function in the corresponding Besov space contains both spiky parts and smooth parts, i.e., the Besov space has inhomogeneous smoothness (Suzuki, 2018).

Our Besov dynamic closure is sufficiently general that subsumes almost all the previous completeness assumptions in the literature. For example, a simple (yet considerably stronger than necessary) sufficient condition for Assumption 5.2 is that the expected reward function $r(s, a)$ and the transition density $P(s'|s, a)$ for each fixed $s'$ are functions in $\bar{B}^\alpha_{p,q}$, regardless of any input function $f$ and any target policy $\pi$. [2] Such a condition on the transition dynamic is common in the RL literature; for example, linear MDPs (Jin et al., 2020a) posit a linear structure on the expected reward and the transition density as $r(s, a) = \langle \phi(s, a), \theta \rangle$ and $P(s'|s, a) = \langle \phi(s, a), \lambda(s') \rangle$ for some feature map $\phi : \mathcal{X} \to \mathbb{R}^{d_0}$ and signed measures $\lambda(s') = (\lambda(s')_1, \ldots, \lambda(s')_{d_0})$. To make it even more concrete, we present the following examples for Assumption 5.2.

**Example 5.2** (*Reproducing kernel Hilbert space (RKHS)*)**.** *Define $k_{h,l}$ the Matérn kernel with smoothness parameter $h > 0$ and length scale $l > 0$. If both $r(\cdot)$ and $g_{s'}(\cdot) := P(s'|\cdot)$ at any $s' \in \mathcal{S}$ are functions in the RKHS of Matérn kernel $k_{h,l}$ where $h = \alpha - d/2 > 0$ and $l > 0$, then Assumption 5.2 holds for $p = q = 2$.* [3] *Moreover, this particular case is equivalent to the dynamic condition considered in Yang et al. (2019).*

**Example 5.3** (Reduction to linear MDPs)**.** *Linear MDPs (Jin et al., 2020a) correspond to Assumption 5.2 with $\alpha = 1$ and $p = q$ on a $p$-norm bounded domain.* [4]

Note that Assumption 5.2 even allows the expected rewards $r(\cdot)$ and the transition densities $g_{s'}(\cdot) := P(s'|\cdot)$ to contain both spiky parts and smooth parts, i.e., inhomogeneous smoothness, as long as $p < 2$ (thus the constraint condition becomes $\alpha > d/p$).

We are now ready to present our main result.

**Theorem 5.1.** *Under Assumption 5.1 and Assumption 5.2 for some $(L, m, S, B)$ satisfying (1), for any $\epsilon > 0, \delta \in (0, 1], K > 0$, if $n$ satisfies that $n \gtrsim \left(\frac{1}{\epsilon^2}\right)^{1+\frac{d}{\alpha}} \log^6 n + \frac{1}{\epsilon^2}(\log(1/\delta) + \log\log n)$, then with probability at least $1 - \delta$, the sup-optimality of Algorithm 1 is*

$$
\begin{cases}
\mathrm{SubOpt}(V_K; \pi) \leq \dfrac{\sqrt{\kappa_\mu}}{1 - \gamma}\epsilon + \dfrac{\gamma^{K/2}}{(1 - \gamma)^{1/2}} & \text{for OPE,} \\[2ex]
\mathrm{SubOpt}(\pi_K) \leq \dfrac{2\gamma\sqrt{\kappa_\mu}}{(1 - \gamma)^2}\epsilon + \dfrac{2\gamma^{1+K/2}}{(1 - \gamma)^{3/2}} & \text{for OPL.}
\end{cases}
$$

*In addition, the optimal deep ReLU network $\Phi(L, m, S, B)$ that obtains such sample complexity (for both OPE and OPL) satisfies*

$$
L \asymp \log N, m \asymp N \log N, S \asymp N, \text{ and } B \asymp N^{1/d + (2\iota)/(\alpha - \iota)}, \tag{1}
$$

*where $N \asymp n^{\frac{1/2 + \left(2 + d^2/(\alpha(\alpha+d))\right)^{-1}}{1 + 2\alpha/d}}$ and $\iota := d(p^{-1} - (1 + \lfloor\alpha\rfloor)^{-1})_+$.*

As the complete form of Theorem 5.1 is quite involved, we interpret and disentangle this result to understand FQI algorithms with neural network function approximation for offline RL tasks. The sub-optimality in both

---

[2]This sufficient condition imposes the smoothness constraint solely on the underlying MDP regardless of the input function $f$. Thus, the "max" over the input function $f(s, a)$ does not affect the smoothness of the resulting function after $f$ is passed through the Bellman operator. This holds regardless of whether $f$ is in the Besov space.

[3]This is due to the norm-equivalence between the above RKHS and the Sobolev space $W_2^\alpha(\mathcal{X})$ (Kanagawa et al., 2018) and the degeneration from Besov spaces to Sobolev spaces as $B_{2,2}^\alpha(\mathcal{X}) = W_2^\alpha(\mathcal{X})$.

[4]However, linear MDPs do not require the smoothness constraint $\alpha > \frac{d}{\min\{p,2\}}$ to ensure a finite Rademacher complexity of linear models. Of course, our analysis addresses significantly more complex and general settings than linear MDPs which we believe is more important than recovering this particular condition of linear MDPs.

OPE and OPL consists of the statistical error (the first term) and the algorithmic error (the second term). While the algorithmic error enjoys the fast linear convergence to $0$ as $K$ gets large, the statistical error reflects the fundamental difficulty of our problems. To make it more interpretable, we present a simplified version of Theorem 5.1 where we state the sample complexity required to obtain a sub-optimality within $\epsilon$.

**Proposition 5.1** (Simplified version of Theorem 5.1). *For any $K \gtrsim H \log(1/\epsilon)$, the sample complexity of Algorithm 1 for OPE Task and OPL Task is $n = \tilde{\mathcal{O}}(H^{2+2\frac{d}{\alpha}} \kappa_\mu^{1+\frac{d}{\alpha}} \epsilon^{-2-2\frac{d}{\alpha}})$ and $n = \tilde{\mathcal{O}}(H^{4+4\frac{d}{\alpha}} \kappa_\mu^{1+\frac{d}{\alpha}} \epsilon^{-2-2\frac{d}{\alpha}})$, respectively. Moreover, the optimal deep ReLU network $\Phi(L, m, S, B)$ for both OPE and OPL Tasks that obtains such sample complexity is $L = \mathcal{O}(\log n), m = \mathcal{O}(n^{2/5} \log n), S = \mathcal{O}(n^{2/5})$, and $\log B = \mathcal{O}(\frac{n^{2/5}}{d})$.*

To discuss our result, we compare it with other existing works in Table 1. As the literature of offline RL is vast, we only compare with representative works of FQI estimators for offline RL with function approximation under a uniform data coverage assumption, as they are directly relevant to our work that uses FQI estimators with neural network function approximation under uniform data coverage. Here, our sample complexity does not scale with the number of states as in tabular MDPs (Yin & Wang, 2020; Yin et al., 2021; Yin & Wang, 2021a) or the inherent Bellman error as in the general function approximation (Munos & Szepesvári, 2008; Le et al., 2019; Duan et al., 2021a). Instead, it explicitly scales with the (possible fractional) smoothness $\alpha$ of the underlying MDP, the dimension $d$ of the input space, the distributional shift measure $\kappa_\mu$ and the effective episode length $H = (1-\gamma)^{-1}$. Importantly, this guarantee is established under the Besov dynamic closure that subsumes the dynamic conditions of the prior results. Compared to Yang et al. (2019), our sample complexity has a strong advantage in long (effective) horizon problems where $H > \frac{d}{\alpha-2d} \log(1/\epsilon)$ [5] and improves it by a factor of $H^{1-2d/\alpha} \epsilon^{-d/\alpha} \log(H^2/\epsilon^2)$. It also suggests that the data splitting in Yang et al. (2019) should be preferred for short (effective) horizon problems. Though our bound has a tighter dependence on $H$ in the long horizon setting, the dependence on $\epsilon$ in our bound is compromised and does not match the minimax rate in the regression setting. We leave as future direction to construct the lower bound for the data-reuse setting of offline RL.

**On the role of deep ReLU networks in offline RL.** We make several remarks about the role of deep networks in offline RL. The role of deep ReLU networks in offline RL is to guarantee a maximal adaptivity to the (spatial) regularity of the functions in Besov space and obtain an optimal approximation error rate that otherwise were not possible with other function approximation such as kernel methods (Suzuki, 2018). Moreover, by the equivalence in the functions that a neural architecture can compute (Yarotsky, 2017), Theorem 5.1 also readily holds for any other continuous piece-wise linear activation functions with finitely many line segments $M$ where the optimal network architecture only increases the number of units and weights by constant factors depending only on $M$. Moreover, we observe that the optimal ReLU network is relatively "thinner" than overparameterized neural networks that have been recently studied in the literature (Arora et al., 2019; Allen-Zhu et al., 2019; Hanin & Nica, 2019; Cao & Gu, 2019; Belkin, 2021) where the width $m$ is a high-order polynomial of $n$. As overparameterization is a key feature for such overparameterized neural networks to obtain a good generalization, it is natural to ask why a thinner neural network in Theorem 5.1 also guarantees a strong generalization for offline RL? Intuitively, the optimal ReLU network in Theorem 5.1 is regularized by a strong sparsity which resonates with our practical wisdom that a sparsity-based regularization prevents over-fitting and achieve a better generalization. Indeed, as the total number of parameters in the considered neural network is $p = md + m + m^2(L-2) = \mathcal{O}(N^2 \log^3 N)$ while the number of non-zeros parameters $S$ only scales with $N$, the optimal ReLU network in Theorem 5.1 is relatively sparse.

## 6 Technical Review

In this section, we highlight the key technical challenges in our analysis. In summary, two key technical challenges in our analysis are rooted in the consideration of the correlated structure in value regression in Algorithm 1, and the use of deep neural network as function approximation (and their combination). To address these challenges, we devise the so-called double uniform convergence argument and leverage a

---

[5] This condition is often easily satisfied as in practice we commonly set $\gamma = 0.99$ and $\epsilon = 0.001$, thus we have $H = 100$ and $\log(1/\epsilon) = 3$.

localization argument via sub-root functions for local Rademacher complexities. In what follows, we briefly discuss these technical challenges and our analysis approach.

The analysis and technical proofs of Yang et al. (2019); Le et al. (2019) heavily rely on the equation $\mathbb{E}\left[r_i + \gamma\mathbb{E}_{a'\sim\pi(\cdot|s_i')}\left[Q_{k-1}(s_i',a)\right]\right] = [T^*Q_{k-1}](s_i,a_i)$ to leverage the standard nonparametric regression techniques (in a supervised learning setting). However, the correlated structure in Algorithm 1 implies $\mathbb{E}\left[r_i + \gamma\mathbb{E}_{a'\sim\pi(\cdot|s_i')}\left[Q_{k-1}(s_i',a)\right]\right] \neq [T^*Q_{k-1}](s_i,a_i)$ as $Q_{k-1}$ also depends on $(s_i,a_i)$. Thus, the techniques in these prior works could not be used here and we require a new analysis. It is worth noting that Le et al. (2019) also re-use the data as in Algorithm 1 (instead of data splitting as in Yang et al. (2019)) but mistakenly assume that $\mathbb{E}\left[r_i + \gamma\mathbb{E}_{a'\sim\pi(\cdot|s_i')}\left[Q_{k-1}(s_i',a)\right]\right] = [T^*Q_{k-1}](s_i,a_i)$. To deal with the correlated structure, we devise a *double* uniform convergence argument. The double uniform convergence argument is appealingly intuitive: while in a standard regression problem, the (single) uniform convergence argument seeks the generalization guarantee uniformly over an entire hypothesis space of a data-dependent empirical risk minimizer, in the value regression problem of Algorithm 1, we additionally guarantee generalization uniformly over the hypothesis space of the data-dependent regression target $T^*Q_{k-1}$. To make it concrete, we highlight a key equality in our proof where the double uniform convergence argument is used:

$$\max_k \|Q_{k+1} - T^*Q_k\|_\mu^2 = \underbrace{\sup_{Q\in\mathcal{F}_{\mathrm{NN}}}(\mathbb{E}-\mathbb{E}_n)(l_{\hat{f}^Q} - l_{f_*^Q})}_{I_1,\text{empirical process term}} + \underbrace{\sup_{Q\in\mathcal{F}_{\mathrm{NN}}}\mathbb{E}_n(l_{f_\perp^Q} - l_{f_*^Q})}_{I_2,\text{bias term}},$$

where $f_*^Q(x) = \mathbb{E}[r + \gamma\max_{a'}Q(s',a')|x]$, and $f_\perp^Q := \arg\inf_{f\in\mathcal{F}_{\mathrm{NN}}}\|f - f_*^Q\|_{2,\mu}$, and $l_{f_\perp^Q} := (f_\perp^Q(x_1) - r_1 - \gamma\max_{a'}Q(s_1',a'))^2$ and $l_{f_*^Q} := (f_*^Q(x_1) - r_1 - \gamma\max_{a'}Q(s_1',a'))^2$ are random variables with respect to the randomness of $(x_1,s_1',r_1)$. We have learned that a similar general idea of the double uniform convergence argument has been leveraged in Chen & Jiang (2019b) for general function classes. We remark they use finite function classes, and in our case, the double uniform convergence argument is particularly helpful in dealing with local Rademacher complexities under a data-dependent structure as local Rademacher complexities already involve the supremum operator which can be naturally incorporated with the double uniform convergence argument.

The double uniform convergence argument also requires a different technique to control an empirical process term $I_1$ as it now requires a uniform convergence over the regression target. We leverage local Rademacher complexities to derive a bound on $I_1$:

$$\sup\{(\mathbb{E}-\mathbb{E}_n)(l_{\hat{f}^Q} - l_{f_*^Q}) : Q\in\mathcal{F}_{\mathrm{NN}}, \|\hat{f}^Q - f_*^Q\|_\mu^2 \leq r\}$$

$$\leq 6\mathbb{E}R_n\left\{f - g : f\in\mathcal{F}_{\mathrm{NN}}, g\in T^*\mathcal{F}_{\mathrm{NN}}, \|f-g\|_\mu^2 \leq r\right\} + 2\sqrt{\frac{2r\log(1/\delta)}{n}} + \frac{28\log(1/\delta)}{3n}.$$

where $R_n$ is the local Rademacher complexity (Bartlett et al., 2005). An explicit bound is then derived via a localization argument and the fixed point of a sub-root function.

The use of neural networks pose a new challenge mainly in bounding the bias term $I_2$. We derive this bound using the adaptivity of deep ReLU network to the regularity in Besov spaces, leveraging our Besov dynamic condition in Assumption 5.2. Bounding the bias term also requires the use of a concentration inequality. While Le et al. (2019) use Bernstein's inequality, our bias term $I_2$ requires a uniform convergence version of Bernstein's inequality as $I_2$ requires a guarantee uniformly over $\mathcal{F}_{\mathrm{NN}}$. We omit a detailed proof for Theorem 5.1 to Section A.

# 7 Conclusion and Discussion

We presented the sample complexity of FQI estimators for offline RL with deep ReLU network function approximation under a uniform data coverage assumption. We proved that the FQI-type algorithm achieved the sample complexity of $n = \tilde{\mathcal{O}}(H^{4+4\frac{d}{\alpha}}\kappa_\mu^{1+\frac{d}{\alpha}}\epsilon^{-2-2\frac{d}{\alpha}})$ under a correlated structure and a general dynamic condition namely the Besov dynamic closure. In addition, we corrected the mistake in ignoring the correlated

structure when reusing data with FQI estimators in Le et al. (2019), avoided the possibly inefficient data splitting technique in Yang et al. (2019) for long (effective) horizon problems, and proposed a general dynamic condition that subsumes all the previous Bellmen completeness assumptions. In the following, we discuss future directions.

**Relaxing the assumption about uniform data coverage.** For a future work, we can include the pessimistic approach in Jin et al. (2020b); Rashidinejad et al. (2021); Uehara & Sun (2021); Nguyen-Tang et al. (2022a) to the current work with a more involved analysis of uncertainty quantifiers under non-linear function approximation to relax the strictness of the uniform data coverage assumption.

**Relaxing the assumption about optimization oracle.** The present work assumes access to the optimization oracle when fitting a neural network to the offline data. It is desirable to understand how optimization and generalization of a trained neural network can contribute to offline RL with neural function approximation. A promising approach to obtain a tight trajectory-dependent sub-optimality bound of offline RL with neural function approximation is to characterize the SGD-based optimization via a stochastic differential equation by allowing the stochastic noises to follow the fractional Brownian motion Tan et al. (2022); Tong et al. (2022).

## Acknowledgements

We thank our anonymous reviewers and our action editor Yu-Xiang Wang (UC Santa Barbara) at TMLR for the constructive comments and feedback. Thanh Nguyen-Tang thank Le Minh Khue Nguyen (University of Rochester) for the support of this project during the COVID times.

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

# A  Appendix

# A  Proof of Theorem 5.1

We now provide a complete proof of Theorem 5.1. The proof has four main components: a sub-optimality decomposition for error propagation across iterations, a Bellman error decomposition using a uniform convergence argument, a deviation analysis for least squares with deep ReLU networks using local Rademacher complexities and a localization argument, and a upper bound minimization step to obtain an optimal deep ReLU architecture.

**Step 1: A sub-optimality decomposition**

The first step of the proof is a sub-optimality decomposition, stated in Lemma A.1, that applies generally to any least-squares Q-iteration methods.

**Lemma A.1** (*A sub-optimality decomposition*). *Under Assumption 5.1, the sub-optimality of $V_K$ returned by Algorithm 1 is bounded as*

$$
\mathrm{SubOpt}(V_K) \leq
\begin{cases}
\frac{\sqrt{\kappa_\mu}}{1-\gamma} \max\limits_{0 \leq k \leq K-1} \|Q_{k+1} - T^\pi Q_k\|_\mu + \frac{\gamma^{K/2}}{(1-\gamma)^{1/2}} & \text{for OPE,} \\
\frac{4\gamma\sqrt{\kappa_\mu}}{(1-\gamma)^2} \max\limits_{0 \leq k \leq K-1} \|Q_{k+1} - T^* Q_k\|_\mu + \frac{4\gamma^{1+K/2}}{(1-\gamma)^{3/2}} & \text{for OPL.}
\end{cases}
$$

The lemma states that the sub-optimality decomposes into a statistical error (the first term) and an algorithmic error (the second term). While the algorithmic error enjoys the fast linear convergence rate, the statistical error arises from the distributional shift in the offline data and the estimation error of the target $Q$-value functions due to finite data. Crucially, the contraction of the (optimality) Bellman operators $T^\pi$ and $T^*$ allows the sup-optimality error at the final iteration $K$ to propagate across all iterations $k \in [0, K-1]$. Note that this result is agnostic to any function approximation form and does not require Assumption 5.2. The result uses a relatively standard argument that appears in a number of works on offline RL (Munos & Szepesvári, 2008; Le et al., 2019).

*Proof of Lemma A.1.* We will prove the sup-optimality decomposition for both settings: OPE and OPL.

**(i) For OPE.** We denote the right-linear operator by $P^\pi \cdot : \{\mathcal{X} \to \mathbb{R}\} \to \{\mathcal{X} \to \mathbb{R}\}$ where

$$
(P^\pi f)(s, a) := \int_\mathcal{X} f(s', a') \pi(da'|s') P(ds'|s, a),
$$

for any $f \in \{\mathcal{X} \to \mathbb{R}\}$. Denote Denote $\rho^\pi(dsda) = \rho(ds)\pi(da|s)$. Let $\epsilon_k := Q_{k+1} - T^\pi Q_k, \forall k \in [0, K-1]$ and $\epsilon_K = Q_0 - Q^\pi$. Since $Q^\pi$ is the (unique) fixed point of $T^\pi$, we have

$$
Q_k - Q^\pi = T^\pi Q_{k-1} - T^\pi Q^\pi + \epsilon_{k-1} = \gamma P^\pi (Q_{k-1} - Q^\pi) + \epsilon_{k-1}.
$$

By recursion, we have

$$
Q_K - Q^\pi = \sum_{k=0}^{K} (\gamma P^\pi)^k \epsilon_k = \frac{1 - \gamma^{K+1}}{1 - \gamma} \sum_{k=0}^{K} \alpha_k A_k \epsilon_k
$$

where $\alpha_k := \frac{(1-\gamma)\gamma^k}{1-\gamma^{K+1}}, \forall k \in [K]$ and $A_k := (P^\pi)^k, \forall k \in [K]$. Note that $\sum_{k=0}^{K} \alpha_k = 1$ and $A_k$'s are probability kernels. Denoting by $|f|$ the point-wise absolute value $|f(s, a)|$, we have that the following inequality holds point-wise:

$$
|Q_K - Q^\pi| \leq \frac{1 - \gamma^{K+1}}{1 - \gamma} \sum_{k=0}^{K} \alpha_k A_k |\epsilon_k|.
$$

We have

$$\|Q_K - Q^\pi\|_{\rho^\pi}^2 \leq \frac{(1-\gamma^{K+1})^2}{(1-\gamma)^2} \int \rho(ds)\pi(da|s) \left( \sum_{k=0}^K \alpha_k A_k |\epsilon_k|(s,a) \right)^2$$

$$\overset{(a)}{\leq} \frac{(1-\gamma^{K+1})^2}{(1-\gamma)^2} \int \rho(ds)\pi(da|s) \sum_{k=0}^K \alpha_k A_k^2 \epsilon_k^2(s,a)$$

$$\overset{(b)}{\leq} \frac{(1-\gamma^{K+1})^2}{(1-\gamma)^2} \int \rho(ds)\pi(da|s) \sum_{k=0}^K \alpha_k A_k \epsilon_k^2(s,a)$$

$$\overset{(c)}{\leq} \frac{(1-\gamma^{K+1})^2}{(1-\gamma)^2} \left( \int \rho(ds)\pi(da|s) \sum_{k=0}^{K-1} \alpha_k A_k \epsilon_k^2(s,a) + \alpha_K \right)$$

$$\overset{(d)}{\leq} \frac{(1-\gamma^{K+1})^2}{(1-\gamma)^2} \left( \int \mu(ds,da) \sum_{k=0}^{K-1} \alpha_k \kappa_\mu \epsilon_k^2(s,a) + \alpha_K \right)$$

$$= \frac{(1-\gamma^{K+1})^2}{(1-\gamma)^2} \left( \sum_{k=0}^{K-1} \alpha_k \kappa_\mu \|\epsilon_k\|_\mu^2 + \alpha_K \right)$$

$$\leq \frac{\kappa_\mu}{(1-\gamma)^2} \max_{0 \leq k \leq K-1} \|\epsilon_k\|_\mu^2 + \frac{\gamma^K}{(1-\gamma)}.$$

The inequalities $(a)$ and $(b)$ follow from Jensen's inequality, $(c)$ follows from $\|Q_0\|_\infty, \|Q^\pi\|_\infty \leq 1$, and $(d)$ follows from Assumption 5.1 that $\rho^\pi A_k = \rho^\pi (P^\pi)^k \leq \kappa_\mu \mu$. Thus we have

$$\mathrm{SubOpt}(V_K; \pi) = |V_K - V^\pi|$$

$$= \left| \mathbb{E}_{\rho,\pi}[Q_K(s,a)] - \mathbb{E}_\rho[Q^\pi(s,a)] \right|$$

$$\leq \mathbb{E}_{\rho,\pi} \left[ |Q_K(s,a) - Q^\pi(s,a)| \right]$$

$$\leq \sqrt{\mathbb{E}_{\rho,\pi} \left[ (Q_K(s,a) - Q^\pi(s,a))^2 \right]}$$

$$= \|Q_K - Q^\pi\|_{\rho^\pi}$$

$$\leq \frac{\sqrt{\kappa_\mu}}{1-\gamma} \max_{0 \leq k \leq K-1} \|\epsilon_k\|_\mu + \frac{\gamma^{K/2}}{(1-\gamma)^{1/2}}.$$

**(ii) For OPL.** The sup-optimality for the OPL setting is more complex than the OPE setting but the technical steps are relatively similar. In particular, let $\epsilon_{k-1} = T^* Q_{k-1} - Q_k, \forall k$ and $\pi^*(s) = \arg\max_a Q^*(s,a), \forall s$, we have

$$Q^* - Q_K = T^{\pi^*} Q^* - T^{\pi^*} Q_{K-1} + \underbrace{T^{\pi^*} Q_{K-1} - T^* Q_{K-1}}_{\leq 0} + \epsilon_{K-1}$$

$$\leq \gamma P^{\pi^*} (Q^* - Q_{K-1}) + \epsilon_{K-1}$$

$$\leq \sum_{k=0}^{K-1} \gamma^{K-k-1} (P^{\pi^*})^{K-k-1} \epsilon_k + \gamma^K (P^{\pi^*})^K (Q^* - Q_0) \text{ (by recursion).} \tag{2}$$

Now, let $\pi_k$ be the greedy policy w.r.t. $Q_k$, we have

$$Q^* - Q_K = \underbrace{T^{\pi^*} Q^*}_{\geq T^{\pi_{K-1}} Q^*} - T^{\pi_{K-1}} Q_{K-1} + \underbrace{T^{\pi_{K-1}} Q_{K-1} - T^* Q_{K-1}}_{\geq 0} + \epsilon_{K-1}$$

$$\geq \gamma P^{\pi_{K-1}} (Q^* - Q_{K-1}) + \epsilon_{K-1}$$

$$\geq \sum_{k=0}^{K-1} \gamma^{K-k-1} (P^{\pi_{K-1}} \ldots P^{\pi_{k+1}}) \epsilon_k + \gamma^K (P^{\pi_{K-1}} \ldots P^{\pi_0})(Q^* - Q_0). \tag{3}$$

Now, we turn to decompose $Q^* - Q^{\pi_K}$ as

$$Q^* - Q^{\pi_K} = (T^{\pi^*}Q^* - T^{\pi^*}Q_K) + \underbrace{(T^{\pi^*}Q_K - T^{\pi_K}Q_K)}_{\leq 0} + (T^{\pi_K}Q_K - T^{\pi_K}Q^{\pi_K})$$

$$\leq \gamma P^{\pi^*}(Q^* - Q_K) + \gamma P^{\pi_K}(Q_K - Q^* + Q^* - Q^{\pi_K}).$$

Thus, we have

$$(I - \gamma P^{\pi_K})(Q^* - Q^{\pi_K}) \leq \gamma(P^{\pi^*} - P^{\pi_K})(Q^* - Q_K).$$

Note that the operator $(I - \gamma P^{\pi_K})^{-1} = \sum_{i=0}^{\infty}(\gamma P^{\pi_K})^i$ is monotone, thus

$$Q^* - Q^{\pi_K} \leq \gamma(I - \gamma P^{\pi_K})^{-1}P^{\pi^*}(Q^* - Q_K) - \gamma(I - \gamma P^{\pi_K})^{-1}P^{\pi_K}(Q^* - Q_K). \tag{4}$$

Combining Equation (4) with Equations (2) and (3), we have

$$Q^* - Q^{\pi_K} \leq (I - \gamma P^{\pi_K})^{-1}\left(\sum_{k=0}^{K-1}\gamma^{K-k}(P^{\pi^*})^{K-k}\epsilon_k + \gamma^{K+1}(P^{\pi^*})^{K+1}(Q^* - Q_0)\right) -$$

$$(I - \gamma P^{\pi_K})^{-1}\left(\sum_{k=0}^{K-1}\gamma^{K-k}(P^{\pi_K}\ldots P^{\pi_{k+1}})\epsilon_k + \gamma^{K+1}(P^{\pi_K}\ldots P^{\pi_0})(Q^* - Q_0)\right).$$

Using the triangle inequality, the above inequality becomes

$$Q^* - Q^{\pi_K} \leq \frac{2\gamma(1 - \gamma^{K+1})}{(1-\gamma)^2}\left(\sum_{k=0}^{K-1}\alpha_k A_k|\epsilon_k| + \alpha_K A_K|Q^* - Q_0|\right),$$

where

$$A_k = \frac{1-\gamma}{2}(I - \gamma P^{\pi_K})^{-1}\left((P^{\pi^*})^{K-k} + P^{\pi_K}\ldots P^{\pi_{k+1}}\right), \forall k < K,$$

$$A_K = \frac{1-\gamma}{2}(I - \gamma P^{\pi_K})^{-1}\left((P^{\pi^*})^{K+1} + P^{\pi_K}\ldots P^{\pi_0}\right),$$

$$\alpha_k = \gamma^{K-k-1}(1-\gamma)/(1-\gamma^{K+1}), \forall k < K,$$

$$\alpha_K = \gamma^K(1-\gamma)/(1-\gamma^{K+1}).$$

Note that $A_k$ is a probability kernel for all $k$ and $\sum_k \alpha_k = 1$. Thus, similar to the steps in the OPE setting, for any policy $\pi$, we have

$$\|Q^* - Q^{\pi_K}\|_{\rho^{\pi}}^2 \leq \left[\frac{2\gamma(1 - \gamma^{K+1})}{(1-\gamma)^2}\right]^2\left(\int \rho(ds)\pi(da|s)\sum_{k=0}^{K-1}\alpha_k A_k \epsilon_k^2(s,a) + \alpha_K\right)$$

$$\leq \left[\frac{2\gamma(1 - \gamma^{K+1})}{(1-\gamma)^2}\right]^2\left(\int \mu(ds,da)\sum_{k=0}^{K-1}\alpha_k \kappa_\mu \epsilon_k^2(s,a) + \alpha_K\right)$$

$$= \left[\frac{2\gamma(1 - \gamma^{K+1})}{(1-\gamma)^2}\right]^2\left(\sum_{k=0}^{K-1}\alpha_k \kappa_\mu \|\epsilon_k\|_\mu^2 + \alpha_K\right)$$

$$\leq \frac{4\gamma^2\kappa_\mu}{(1-\gamma)^4}\max_{0\leq k\leq K-1}\|\epsilon_k\|_\mu^2 + \frac{4\gamma^{K+2}}{(1-\gamma)^3}.$$

Thus, we have

$$\text{SubOpt}(\pi^K) = \|Q^* - Q^{\pi_K}\|_{\rho^{\pi}} \leq \frac{2\gamma\sqrt{\kappa_\mu}}{(1-\gamma)^2}\max_{0\leq k\leq K-1}\|\epsilon_k\|_\mu + \frac{2\gamma^{K/2+1}}{(1-\gamma)^{3/2}}.$$

$$\square$$

**Step 2: A Bellman error decomposition**

The next step of the proof is to decompose the Bellman errors $\|Q_{k+1} - T^\pi Q_k\|_\mu$ for OPE and $\|Q_{k+1} - T^* Q_k\|_\mu$ for OPL. Since these errors can be decomposed and bounded similarly, we only focus on OPL here.

The difficulty in controlling the estimation error $\|Q_{k+1} - T^* Q_k\|_{2,\mu}$ is that $Q_k$ itself is a random variable that depends on the offline data $\mathcal{D}$. In particular, at any fixed $k$ with Bellman targets $\{y_i\}_{i=1}^n$ where $y_i = r_i + \gamma \max_{a'} Q_k(s_i', a')$, it is not immediate that $\mathbb{E}\left[[T^* Q_k](x_i) - y_i | x_i\right] = 0$ for each covariate $x_i := (s_i, a_i)$ as $Q_k$ itself depends on $x_i$ (thus the tower law cannot apply here). A naive and simple approach to break such data dependency of $Q_k$ is to split the original data $\mathcal{D}$ into $K$ disjoint subsets and estimate each $Q_k$ using a separate subset. This naive approach is equivalent to the setting in Yang et al. (2019) where a fresh batch of data is generated for different iterations. This approach is however not efficient as it uses only $n/K$ samples to estimate each $Q_k$. This is problematic in high-dimensional offline RL when the number of iterations $K$ can be very large as it is often the case in practical settings. We instead prefer to use all $n$ samples to estimate each $Q_k$. This requires a different approach to handle the complicated data dependency of each $Q_k$. To circumvent this issue, we leverage a uniform convergence argument by introducing a deterministic covering of $T^* \mathcal{F}_{\mathrm{NN}}$. Each element of the deterministic covering induces a different regression target $\{r_i + \gamma \max_{a'} \tilde{Q}(s_i', a')\}_{i=1}^n$ where $\tilde{Q}$ is a deterministic function from the covering which ensures that $\mathbb{E}\left[r_i + \gamma \max_{a'} \tilde{Q}(s_i', a') - [T^* \tilde{Q}](x_i) | x_i\right] = 0$. In particular, we denote

$$y_i^{Q_k} = r_i + \gamma \max_{a'} Q_k(s_i', a'), \forall i \text{ and } \hat{f}^{Q_k} := Q_{k+1} = \arg\inf_{f \in \mathcal{F}_{\mathrm{NN}}} \sum_{i=1}^n l(f(x_i), y_i^{Q_k}), \text{ and } f_*^{Q_k} = T^* Q_k,$$

where $l(x, y) = (x - y)^2$ is the squared loss function. Note that for any deterministic $Q \in \mathcal{F}_{\mathrm{NN}}$, we have $f_*^Q(x_1) = \mathbb{E}[y_1^Q | x_1], \forall x_1$, thus

$$\mathbb{E}(l_f - l_{f_*^Q}) = \|f - f_*^Q\|_\mu^2, \forall f, \tag{5}$$

where $l_f$ denotes the random variable $(f(x_1) - y_1^Q)^2$ for a given fixed $Q$. Now letting $f_\perp^Q := \arg\inf_{f \in \mathcal{F}_{\mathrm{NN}}} \|f - f_*^Q\|_{2,\mu}$ be the projection of $f_*^Q$ onto the function class $\mathcal{F}_{\mathrm{NN}}$, we have

$$\max_k \|Q_{k+1} - T^* Q_k\|_\mu^2 = \max_k \|\hat{f}^{Q_k} - f_*^{Q_k}\|_\mu^2 \overset{(a)}{\leq} \sup_{Q \in \mathcal{F}_{\mathrm{NN}}} \|\hat{f}^Q - f_*^Q\|_\mu^2 \overset{(b)}{=} \sup_{Q \in \mathcal{F}_{\mathrm{NN}}} \mathbb{E}(l_{\hat{f}^Q} - l_{f_*^Q})$$

$$\overset{(c)}{\leq} \sup_{Q \in \mathcal{F}_{\mathrm{NN}}} \left\{ \mathbb{E}(l_{\hat{f}^Q} - l_{f_*^Q}) + \mathbb{E}_n(l_{f_\perp^Q} - l_{\hat{f}^Q}) \right\}$$

$$= \sup_{Q \in \mathcal{F}_{\mathrm{NN}}} \left\{ (\mathbb{E} - \mathbb{E}_n)(l_{\hat{f}^Q} - l_{f_*^Q}) + \mathbb{E}_n(l_{f_\perp^Q} - l_{f_*^Q}) \right\}$$

$$\leq \underbrace{\sup_{Q \in \mathcal{F}_{\mathrm{NN}}} (\mathbb{E} - \mathbb{E}_n)(l_{\hat{f}^Q} - l_{f_*^Q})}_{I_1, \text{empirical process term}} + \underbrace{\sup_{Q \in \mathcal{F}_{\mathrm{NN}}} \mathbb{E}_n(l_{f_\perp^Q} - l_{f_*^Q})}_{I_2, \text{bias term}}, \tag{6}$$

where (a) follows from that $Q_k \in \mathcal{F}_{\mathrm{NN}}$, (b) follows from Equation (5), and (c) follows from that $\mathbb{E}_n[l_{\hat{f}^Q}] \leq \mathbb{E}_n[l_{f^Q}], \forall f, Q \in \mathcal{F}_{\mathrm{NN}}$. That is, the error is decomposed into two terms: the first term $I_1$ resembles the empirical process in statistical learning theory and the second term $I_2$ specifies the bias caused by the regression target $f_*^Q$ not being in the function space $\mathcal{F}_{\mathrm{NN}}$.

**Step 3: A deviation analysis**

The next step is to bound the empirical process term and the bias term via an intricate concentration, local Rademacher complexities and a localization argument. First, the bias term in Equation (6) is taken uniformly over the function space, thus standard concentration arguments such as Bernstein's inequality and Pollard's inequality used in Munos & Szepesvári (2008); Le et al. (2019) do not apply here. Second, local Rademacher complexities (Bartlett et al., 2005) are data-dependent complexity measures that exploit the fact that only a small subset of the function class will be used. Leveraging a localization argument for

local Rademacher complexities (Farrell et al., 2018), we localize an empirical Rademacher ball into smaller balls by which we can handle their complexities more effectively. Moreover, we explicitly use the sub-root function argument to derive our bound and extend the technique to the uniform convergence case. That is, reasoning over the sub-root function argument makes our proof more modular and easier to incorporate the uniform convergence argument.

Localization is particularly useful to handle the complicated approximation errors induced by deep ReLU network function approximation.

**Step 3.a: Bounding the bias term via a uniform convergence concentration inequality**

Before delving into our proof, we introduce relevant notations. Let $\mathcal{F} - \mathcal{G} := \{f - g : f \in \mathcal{F}, g \in \mathcal{G}\}$, let $N(\epsilon, \mathcal{F}, \|\cdot\|)$ be the $\epsilon$-covering number of $\mathcal{F}$ w.r.t. $\|\cdot\|$ norm, $H(\epsilon, \mathcal{F}, \|\cdot\|) := \log N(\epsilon, \mathcal{F}, \|\cdot\|)$ be the entropic number, let $N_{[]}(\epsilon, \mathcal{F}, \|\cdot\|)$ be the bracketing number of $\mathcal{F}$, i.e., the minimum number of brackets of $\|\cdot\|$-size less than or equal to $\epsilon$, necessary to cover $\mathcal{F}$, let $H_{[]}(\epsilon, \mathcal{F}, \|\cdot\|) = \log N_{[]}(\epsilon, \mathcal{F}, \|\cdot\|)$ be the $\|\cdot\|$-bracketing metric entropy of $\mathcal{F}$, let $\mathcal{F}|\{x_i\}_{i=1}^n = \{(f(x_1), ..., f(x_n)) \in \mathbb{R}^n | f \in \mathcal{F}\}$, and let $T^*\mathcal{F} = \{T^*f : f \in \mathcal{F}\}$. Finally, for sample set $\{x_i\}_{i=1}^n$, we define the empirical norm $\|f\|_n := \sqrt{\frac{1}{n} \sum_{i=1}^n f(x_i)^2}$.

We define the inherent Bellman error as $d_{\mathcal{F}_{\mathrm{NN}}} := \sup_{Q \in \mathcal{F}_{\mathrm{NN}}} \inf_{f \in \mathcal{F}_{\mathrm{NN}}} \|f - T^*Q\|_\mu$. This implies that

$$d_{\mathcal{F}_{\mathrm{NN}}}^2 := \sup_{Q \in \mathcal{F}_{\mathrm{NN}}} \inf_{f \in \mathcal{F}_{\mathrm{NN}}} \|f - T^*Q\|_\mu^2 = \sup_{Q \in \mathcal{F}_{\mathrm{NN}}} \mathbb{E}(l_{f_\perp^Q} - l_{f_*^Q}). \tag{7}$$

We have

$$|l_f - l_g| \le 4|f - g| \text{ and } |l_f - l_g| \le 8.$$

We have

$$H(\epsilon, \{l_{f_\perp^Q} - l_{f_*^Q} : Q \in \mathcal{F}_{\mathrm{NN}}\}|\{x_i, y_i\}_{i=1}^n, n^{-1}\|\cdot\|_1)$$
$$\le H(\frac{\epsilon}{4}, \{f_\perp^Q - f_*^Q : Q \in \mathcal{F}_{\mathrm{NN}}\}|\{x_i\}_{i=1}^n, n^{-1}\|\cdot\|_1)$$
$$\le H(\frac{\epsilon}{4}, (\mathcal{F} - T^*\mathcal{F}_{\mathrm{NN}})|\{x_i\}_{i=1}^n, n^{-1}\|\cdot\|_1)$$
$$\le H(\frac{\epsilon}{8}, \mathcal{F}_{\mathrm{NN}}|\{x_i\}_{i=1}^n, n^{-1}\|\cdot\|_1) + H(\frac{\epsilon}{8}, T^*\mathcal{F}_{\mathrm{NN}}|\{x_i\}_{i=1}^n, n^{-1}\|\cdot\|_1)$$
$$\le H(\frac{\epsilon}{8}, \mathcal{F}_{\mathrm{NN}}|\{x_i\}_{i=1}^n, \|\cdot\|_\infty) + H(\frac{\epsilon}{8}, T^*\mathcal{F}_{\mathrm{NN}}, \|\cdot\|_\infty)$$

For any $\epsilon' > 0$ and $\delta' \in (0, 1)$, it follows from Lemma B.2 with $\epsilon = 1/2$ and $\alpha = \epsilon'^2$, with probability at least $1 - \delta'$, for any $Q \in \mathcal{F}_{\mathrm{NN}}$, we have

$$\mathbb{E}_n(l_{f_\perp^Q} - l_{f_*^Q}) \le 3\mathbb{E}(l_{f_\perp^Q} - l_{f_*^Q}) + \epsilon'^2 \le 3d_{\mathcal{F}_{\mathrm{NN}}}^2 + \epsilon'^2, \tag{8}$$

given that

$$n \approx \frac{1}{\epsilon'^2}\left(\log(4/\delta') + \log \mathbb{E}N(\frac{\epsilon'^2}{40}, (\mathcal{F}_{\mathrm{NN}} - T^*\mathcal{F}_{\mathrm{NN}})|\{x_i\}_{i=1}^n, n^{-1}\|\cdot\|_1)\right).$$

Note that if we use Pollard's inequality (Munos & Szepesvári, 2008) in the place of Lemma B.2, the RHS of Equation (8) is bounded by $\epsilon'$ instead of $\epsilon'^2$ (i.e., $n$ scales with $O(1/\epsilon'^4)$ instead of $O(1/\epsilon'^2)$). In addition, unlike Le et al. (2019), the uniform convergence argument hinders the application of Bernstein's inequality. We remark that Le et al. 2019 makes a mistake in their proof by ignoring the data-dependent structure in the algorithm (i.e., they wrongly assume that $Q^k$ in Algorithm 1 is fixed and independent of $\{s_i, a_i\}_{i=1}^n$). Thus, the uniform convergence argument in our proof is necessary.

**Step 3.b: Bounding the empirical process term via local Rademacher complexities**

For any $Q \in \mathcal{F}_{\mathrm{NN}}$, we have

$$|l_{f_\perp^Q} - l_{f_*^Q}| \leq 2|f_\perp^Q - f_*^Q| \leq 2,$$

$$\mathbb{V}[l_{f_\perp^Q} - l_{f_*^Q}] \leq \mathbb{E}[(l_{f_\perp^Q} - l_{f_*^Q})^2] \leq 4\mathbb{E}(f_\perp^Q - f_*^Q)^2.$$

Thus, it follows from Lemma 1 (with $\alpha = 1/2$) that with any $r > 0, \delta \in (0, 1)$, with probability at least $1 - \delta$, we have

$$\sup\{(\mathbb{E} - \mathbb{E}_n)(l_{\hat{f}^Q} - l_{f_*^Q}) : Q \in \mathcal{F}_{\mathrm{NN}}, \|\hat{f}^Q - f_*^Q\|_\mu^2 \leq r\}$$

$$\leq \sup\{(\mathbb{E} - \mathbb{E}_n)(l_f - l_g) : f \in \mathcal{F}_{\mathrm{NN}}, g \in T^*\mathcal{F}, \|f - g\|_\mu^2 \leq r\}$$

$$\leq 3\mathbb{E}R_n\left\{l_f - l_g : f \in \mathcal{F}_{\mathrm{NN}}, g \in T^*\mathcal{F}_{\mathrm{NN}}, \|f - g\|_\mu^2 \leq r\right\} + 2\sqrt{\frac{2r\log(1/\delta)}{n}} + \frac{28\log(1/\delta)}{3n}$$

$$\leq 6\mathbb{E}R_n\left\{f - g : f \in \mathcal{F}_{\mathrm{NN}}, g \in T^*\mathcal{F}_{\mathrm{NN}}, \|f - g\|_\mu^2 \leq r\right\} + 2\sqrt{\frac{2r\log(1/\delta)}{n}} + \frac{28\log(1/\delta)}{3n}.$$

**Step 3.c: Bounding $\|Q_{k+1} - T^*Q_k\|_\mu$ using localization argument via sub-root functions**

We bound $\|Q_{k+1} - T^*Q_k\|_\mu$ using the localization argument, breaking down the Rademacher complexities into local balls and then build up the original function space from the local balls. Let $\psi$ be a sub-root function (Bartlett et al., 2005, Definition 3.1) with the fixed point $r_*$ and assume that for any $r \geq r_*$, we have

$$\psi(r) \geq 3\mathbb{E}R_n\left\{f - g : f \in \mathcal{F}_{\mathrm{NN}}, g \in T^*\mathcal{F}_{\mathrm{NN}}, \|f - g\|_\mu^2 \leq r\right\}. \tag{9}$$

We recall that a function $\psi : [0, \infty) \to [0, \infty)$ is *sub-root* if it is non-negative, non-decreasing and $r \mapsto \psi(r)/\sqrt{r}$ is non-increasing for $r > 0$. Consequently, a sub-root function $\psi$ has a unique fixed point $r_*$ where $r_* = \psi(r_*)$. In addition, $\psi(r) \leq \sqrt{rr_*}, \forall r \geq r_*$. In the next step, we will find a sub-root function $\psi$ that satisfies the inequality above, but for this step we just assume that we have such $\psi$ at hand. Combining Equations (6), (8), and (9), we have: for any $r \geq r_*$ and any $\delta \in (0, 1)$, if $\|\hat{f}^{Q_{k-1}} - f_*^{Q_{k-1}}\|_{2,\mu}^2 \leq r$, with probability at least $1 - \delta$,

$$\|\hat{f}^{Q_{k-1}} - f_*^{Q_{k-1}}\|_{2,\mu}^2 \leq 2\psi(r) + 2\sqrt{\frac{2r\log(2/\delta)}{n}} + \frac{28\log(2/\delta)}{3n} + 3d_\mathcal{F}^2 + \epsilon'^2$$

$$\leq \sqrt{rr_*} + 2\sqrt{\frac{2r\log(2/\delta)}{n}} + \frac{28\log(2/\delta)}{3n} + (\sqrt{3}d_\mathcal{F} + \epsilon')^2,$$

where

$$n \approx \frac{1}{4\epsilon'^2}\left(\log(8/\delta) + \log \mathbb{E}N(\frac{\epsilon'^2}{20}, (\mathcal{F}_{\mathrm{NN}} - T^*\mathcal{F}_{\mathrm{NN}})|\{x_i\}_{i=1}^n, n^{-1}\|\cdot\|_1)\right).$$

Consider $r_0 \geq r_*$ (to be chosen later) and denote the events

$$B_k := \{\|\hat{f}^{Q_{k-1}} - f_*^{Q_{k-1}}\|_{2,\mu}^2 \leq 2^k r_0\}, \forall k \in \{0, 1, ..., l\},$$

where $l = \log_2(\frac{1}{r_0}) \leq \log_2(\frac{1}{r_*})$. We have $B_0 \subseteq B_1 \subseteq ... \subseteq B_l$ and since $\|f - g\|_\mu^2 \leq 1, \forall |f|_\infty, |g|_\infty \leq 1$, we have $P(B_l) = 1$. If $\|\hat{f}^{Q_{k-1}} - f_*^{Q_{k-1}}\|_\mu^2 \leq 2^i r_0$ for some $i \leq l$, then with probability at least $1 - \delta$, we have

$$\|\hat{f}^{Q_{k-1}} - f_*^{Q_{k-1}}\|_{2,\mu}^2 \leq \sqrt{2^i r_0 r_*} + 2\sqrt{\frac{2^{i+1}r_0 \log(2/\delta)}{n}} + \frac{28\log(2/\delta)}{3n} + (\sqrt{3}d_{\mathcal{F}_{\mathrm{NN}}} + \epsilon')^2$$

$$\leq 2^{i-1}r_0,$$

if the following inequalities hold

$$\sqrt{2^i r_*} + 2\sqrt{\frac{2^{i+1}\log(2/\delta)}{n}} \le \frac{1}{2}2^{i-1}\sqrt{r_0},$$

$$\frac{28\log(2/\delta)}{3n} + (\sqrt{3}d_{\mathcal{F}_{\text{NN}}} + \epsilon')^2 \le \frac{1}{2}2^{i-1}r_0.$$

We choose $r_0 \ge r_*$ such that the inequalities above hold for all $0 \le i \le l$. This can be done by simply setting

$$\sqrt{r_0} = \frac{2}{2^{i-1}}\left(\sqrt{2^i r_*} + 2\sqrt{\frac{2^{i+1}\log(2/\delta)}{n}}\right)\Bigg|_{i=0} + \sqrt{\frac{2}{2^{i-1}}\left(\frac{28\log(2/\delta)}{3n} + (\sqrt{3}d_{\mathcal{F}_{\text{NN}}} + \epsilon')^2\right)}\Bigg|_{i=0}$$

$$\lesssim d_{\mathcal{F}_{\text{NN}}} + \epsilon' + \sqrt{\frac{\log(2/\delta)}{n}} + \sqrt{r_*}.$$

Since $\{B_i\}$ is a sequence of increasing events, we have

$$P(B_0) = P(B_1) - P(B_1 \cap B_0^c) = P(B_2) - P(B_2 \cap B_1^c) - P(B_1 \cap B_0^c)$$

$$= P(B_l) - \sum_{i=0}^{l-1} P(B_{i+1} \cap B_i^c) \ge 1 - l\delta.$$

Thus, with probability at least $1 - \delta$, we have

$$\|\hat{f}^{Q_{k-1}} - f_*^{Q_{k-1}}\|_\mu \lesssim d_{\mathcal{F}_{\text{NN}}} + \epsilon' + \sqrt{\frac{\log(2l/\delta)}{n}} + \sqrt{r_*} \tag{10}$$

where

$$n \approx \frac{1}{4\epsilon'^2}\left(\log(8l/\delta) + \log \mathbb{E}N(\frac{\epsilon'^2}{20}, (\mathcal{F}_{\text{NN}} - T^*\mathcal{F}_{\text{NN}})|\{x_i\}_{i=1}^n, n^{-1}\|\cdot\|_1))\right).$$

### Step 3.d: Finding a sub-root function and its fixed point

It remains to find a sub-root function $\psi(r)$ that satisfies Equation (9) and thus its fixed point. The main idea is to bound the RHS, the local Rademacher complexity, of Equation (9) by its empirical counterpart as the latter can then be further bounded by a sub-root function represented by a measure of compactness of the function spaces $\mathcal{F}_{\text{NN}}$ and $T^*\mathcal{F}_{\text{NN}}$.

For any $\epsilon > 0$, we have the following inequalities for entropic numbers:

$$H(\epsilon, \mathcal{F}_{\text{NN}} - T^*\mathcal{F}_{\text{NN}}, \|\cdot\|_n) \le H(\epsilon/2, \mathcal{F}_{\text{NN}}, \|\cdot\|_n) + H(\epsilon/2, T^*\mathcal{F}_{\text{NN}}, \|\cdot\|_n),$$

$$H(\epsilon, \mathcal{F}_{\text{NN}}, \|\cdot\|_n) \le H(\epsilon, \mathcal{F}_{\text{NN}}|\{x_i\}_{i=1}^n, \|\cdot\|_\infty) \overset{(a)}{\lesssim} N[(\log N)^2 + \log(1/\epsilon)], \tag{11}$$

$$H(\epsilon, T^*\mathcal{F}_{\text{NN}}, \|\cdot\|_n) \le H(\epsilon, T^*\mathcal{F}_{\text{NN}}, \|\cdot\|_\infty) \le H_{[]}(2\epsilon, T^*\mathcal{F}_{\text{NN}}, \|\cdot\|_\infty)$$

$$\overset{(b)}{\le} H_{[]}(2\epsilon, \bar{B}_{p,q}^\alpha(\mathcal{X}), \|\cdot\|_\infty) \overset{(c)}{\lesssim} (2\epsilon)^{-d/\alpha}, \tag{12}$$

where $N$ is a hyperparameter of the deep ReLU network described in Lemma B.9, (a) follows from Lemma B.9, and (b) follows from Assumption 5.2, and (c) follows from Lemma B.8. Let $\mathcal{H} := \mathcal{F}_{\text{NN}} - T^*\mathcal{F}_{\text{NN}}$, it

follows from Lemma B.5 with $\{\xi_k := \epsilon/2^k\}_{k\in\mathbb{N}}$ for any $\epsilon > 0$ that

$$\mathbb{E}_\sigma R_n\{h \in \mathcal{H} - \mathcal{H} : \|h\|_n \le \epsilon\} \le 4 \sum_{k=1}^\infty \frac{\epsilon}{2^{k-1}} \sqrt{\frac{H(\epsilon/2^{k-1}, \mathcal{H}, \|\cdot\|_n)}{n}}$$

$$\le 4 \sum_{k=1}^\infty \frac{\epsilon}{2^{k-1}} \sqrt{\frac{H(\epsilon/2^k, \mathcal{F}_{\mathrm{NN}}, \|\cdot\|_\infty)}{n}} + 4 \sum_{k=1}^\infty \frac{\epsilon}{2^{k-1}} \sqrt{\frac{H(\epsilon/2^k, T^\pi \mathcal{F}_{\mathrm{NN}}, \|\cdot\|_\infty)}{n}}$$

$$\le \frac{4\epsilon}{\sqrt{n}} \sum_{k=1}^\infty 2^{-(k-1)} \sqrt{N\left((\log N)^2 + \log(2^k/\epsilon)\right)} + \frac{4\epsilon}{\sqrt{n}} \sum_{k=1}^\infty 2^{-(k-1)} \sqrt{\left(\frac{\epsilon}{2^{k-1}}\right)^{-d/\alpha}}$$

$$\lesssim \frac{\epsilon}{\sqrt{n}} \sqrt{N((\log N)^2 + \log(1/\epsilon))} + \frac{\epsilon^{1-\frac{d}{2\alpha}}}{\sqrt{n}},$$

where we use $\sqrt{a+b} \le \sqrt{a} + \sqrt{b}, \forall a, b \ge 0$, $\sum_{k=1}^\infty \frac{\sqrt{k}}{2^{k-1}} < \infty$, and $\sum_{k=1}^\infty \left(\frac{1}{2^{1-\frac{d}{2\alpha}}}\right)^{k-1} < \infty$.

It now follows from Lemma B.4 that

$$\mathbb{E}_\sigma R_n\{f \in \mathcal{F}, g \in T^* \mathcal{F} : \|f - g\|_n^2 \le r\}$$

$$\le \inf_{\epsilon>0} \left[ \mathbb{E}_\sigma R_n\{h \in \mathcal{H} - \mathcal{H} : \|h\|_\mu \le \epsilon\} + \sqrt{\frac{2rH(\epsilon/2, \mathcal{H}, \|\cdot\|_n)}{n}} \right]$$

$$\lesssim \left[ \frac{\epsilon}{\sqrt{n}} \sqrt{N((\log N)^2 + \log(1/\epsilon))} + \frac{\epsilon^{1-\frac{d}{2\alpha}}}{\sqrt{n}} + \sqrt{\frac{2r}{n}} \sqrt{N((\log N)^2 + \log(4/\epsilon))} + \sqrt{\frac{2r}{n}} (\epsilon/2)^{\frac{-d}{2\alpha}} \right] \Bigg|_{\epsilon=n^{-\beta}}$$

$$\asymp n^{-\beta-1/2} \sqrt{N(\log^2 N + \log n)} + n^{-\beta(1-\frac{d}{2\alpha})-1/2} + \sqrt{\frac{r}{n}} \sqrt{N(\log^2 N + \log n)} + \sqrt{r} n^{-\frac{1}{2}(1-\frac{\beta d}{\alpha})} =: \psi_1(r),$$

where $\beta \in (0, \frac{\alpha}{d})$ is an absolute constant to be chosen later.

Note that $\mathbb{V}[(f - g)^2] \le \mathbb{E}[(f - g)^4] \le \mathbb{E}[(f - g)^2]$ for any $f \in \mathcal{F}_{\mathrm{NN}}, g \in T^* \mathcal{F}_{\mathrm{NN}}$. Thus, for any $r \ge r_*$, it follows from Lemma B.1 that with probability at least $1 - \frac{1}{n}$, we have the following inequality for any $f \in \mathcal{F}_{\mathrm{NN}}, g \in T^* \mathcal{F}_{\mathrm{NN}}$ such that $\|f - g\|_\mu^2 \le r$,

$$\|f - g\|_n^2$$

$$\le \|f - g\|_\mu^2 + 3\mathbb{E}R_n\{(f - g)^2 : f \in \mathcal{F}_{\mathrm{NN}}, g \in T^* \mathcal{F}_{\mathrm{NN}}, \|f - g\|_\mu^2 \le r\} + \sqrt{\frac{2r \log n}{n}} + \frac{56}{3} \frac{\log n}{n}$$

$$\le \|f - g\|_\mu^2 + 3\mathbb{E}R_n\{f - g : f \in \mathcal{F}_{\mathrm{NN}}, g \in T^* \mathcal{F}_{\mathrm{NN}}, \|f - g\|_\mu^2 \le r\} + \sqrt{\frac{2r \log n}{n}} + \frac{56}{3} \frac{\log n}{n}$$

$$\le r + \psi(r) + r + r \le 4r,$$

if $r \ge r_* \vee \frac{2\log n}{n} \vee \frac{56 \log n}{3n}$. For such $r$, denote $E_r = \{\|f - g\|_n^2 \le 4r\} \cap \{\|f - f_*\|_\mu^2 \le r\}$, we have $P(E_r) \ge 1 - 1/n$ and

$$3\mathbb{E}R_n\{f - g : f \in \mathcal{F}_{\mathrm{NN}}, g \in T^* \mathcal{F}_{\mathrm{NN}}, \|f - g\|_\mu^2 \le r\}$$

$$= 3\mathbb{E}\mathbb{E}_\sigma R_n\{f - g : f \in \mathcal{F}_{\mathrm{NN}}, g \in T^* \mathcal{F}_{\mathrm{NN}}, \|f - g\|_\mu^2 \le r\}$$

$$\le 3\mathbb{E}\left[ 1_{E_r} \mathbb{E}_\sigma R_n\{f - g : f \in \mathcal{F}_{\mathrm{NN}}, g \in T^* \mathcal{F}_{\mathrm{NN}}, \|f - g\|_\mu^2 \le r\} + (1 - 1_{E_r}) \right]$$

$$\le 3\mathbb{E}\left[ \mathbb{E}_\sigma R_n\{f - g : f \in \mathcal{F}_{\mathrm{NN}}, g \in T^* \mathcal{F}_{\mathrm{NN}}, \|f - g\|_n^2 \le 4r\} + (1 - 1_{E_r}) \right]$$

$$\le 3(\psi_1(4r) + \frac{1}{n})$$

$$\lesssim n^{-\beta-1/2} \sqrt{N(\log^2 N + \log n)} + n^{-\beta(1-\frac{d}{2\alpha})-1/2} + \sqrt{\frac{r}{n}} \sqrt{N(\log^2 N + \log n)}$$

$$+ \sqrt{r} n^{-\frac{1}{2}(1-\frac{\beta d}{\alpha})} + n^{-1} =: \psi(r).$$

It is easy to verify that $\psi(r)$ defined above is a sub-root function. The fixed point $r_*$ of $\psi(r)$ can be solved analytically via the simple quadratic equation $r_* = \psi(r_*)$. In particular, we have

$$
\begin{aligned}
\sqrt{r_*} &\lesssim n^{-1/2}\sqrt{N(\log^2 N + \log n)} + n^{-\frac{1}{2}(1-\frac{\beta d}{\alpha})} + n^{-\frac{\beta}{2}-\frac{1}{4}}[N(\log^2 N + \log n)]^{1/4} \\
&\quad + n^{-\frac{\beta}{2}(1-\frac{d}{2\alpha})-\frac{1}{2}} + n^{-1/2} \\
&\lesssim n^{-\frac{1}{4}((2\beta)\wedge 1)+1)}\sqrt{N(\log^2 N + \log n)} + n^{-\frac{1}{2}(1-\frac{\beta d}{\alpha})} + n^{-\frac{\beta}{2}(1-\frac{d}{2\alpha})-\frac{1}{2}} + n^{-1/2}.
\end{aligned}
\tag{13}
$$

It follows from Equation (10) (where $l \lesssim \log(1/r_*)$), the definition of $d_{\mathcal{F}_{\mathrm{NN}}}$, Lemma B.9, and Equation (13) that for any $\epsilon' > 0$ and $\delta \in (0,1)$, with probability at least $1 - \delta$, we have

$$
\begin{aligned}
\max_k \|Q_{k+1} - T^*Q_k\|_\mu &\lesssim N^{-\alpha/d} + \epsilon' + n^{-\frac{1}{4}((2\beta)\wedge 1)+1)}\sqrt{N(\log^2 N + \log n)} + n^{-\frac{1}{2}(1-\frac{\beta d}{\alpha})} \\
&\quad + n^{-\frac{\beta}{2}(1-\frac{d}{2\alpha})-\frac{1}{2}} + n^{-1/2}\sqrt{\log(1/\delta) + \log\log n},
\end{aligned}
\tag{14}
$$

where

$$
n \gtrsim \frac{1}{4\epsilon'^2}\left(\log(1/\delta) + \log\log n + \log \mathbb{E}N(\frac{\epsilon'^2}{20}, (\mathcal{F}_{\mathrm{NN}} - T^*\mathcal{F}_{\mathrm{NN}})|\{x_i\}_{i=1}^n, n^{-1}\cdot\|\cdot\|_1))\right).
\tag{15}
$$

**Step 4: Minimizing the upper bound**

The final step for the proof is to minimize the upper error bound obtained in the previous steps w.r.t. two free parameters $\beta \in (0, \frac{\alpha}{d})$ and $N \in \mathbb{N}$. Note that $N$ parameterizes the deep ReLU architecture $\Phi(L, m, S, B)$ given Lemma B.9. In particular, we optimize over $\beta \in (0, \frac{\alpha}{d})$ and $N \in \mathbb{N}$ to minimize the upper bound in the RHS of Equation (14). The RHS of Equation (14) is minimized (up to $\log n$-factor) by choosing

$$
N \asymp n^{\frac{1}{2}((2\beta)\wedge 1)+1)\frac{d}{2\alpha+d}} \text{ and } \beta = \left(2 + \frac{d^2}{\alpha(\alpha+d)}\right)^{-1},
\tag{16}
$$

which results in $N \asymp n^{\frac{1}{2}(2\beta+1)\frac{d}{2\alpha+d}}$. At these optimal values, Equation (14) becomes

$$
\max_k \|Q_{k+1} - T^*Q_k\|_\mu \lesssim \epsilon' + n^{-\frac{1}{2}\left(\frac{2\alpha}{2\alpha+d}+\frac{d}{\alpha}\right)^{-1}}\log n + n^{-1/2}\sqrt{\log(1/\delta) + \log\log n},
\tag{17}
$$

where we use inequalities $n^{-\frac{\beta}{2}(1-\frac{d}{2\alpha})-\frac{1}{2}} \leq n^{-\frac{1}{2}(1-\frac{\beta d}{\alpha})} \asymp N^{-\alpha/d} = n^{-\frac{1}{2}\left(\frac{2\alpha}{2\alpha+d}+\frac{d}{\alpha}\right)^{-1}}$.

Now, for any $\epsilon > 0$, we set $\epsilon' = \epsilon/3$ and let

$$
n^{-\frac{1}{2}\left(\frac{2\alpha}{2\alpha+d}+\frac{d}{\alpha}\right)^{-1}}\log n \lesssim \epsilon/3 \text{ and } n^{-1/2}\sqrt{\log(1/\delta) + \log\log n} \lesssim \epsilon/3.
$$

It then follows from Equation (17) that with probability at least $1 - \delta$, we have $\max_k \|Q_{k+1} - T^*Q_k\|_\mu \leq \epsilon$ if $n$ simultaneously satisfies Equation (15) with $\epsilon' = \epsilon/3$ and

$$
n \gtrsim \left(\frac{1}{\epsilon^2}\right)^{\frac{2\alpha}{2\alpha+d}+\frac{d}{\alpha}}(\log^2 n)^{\frac{2\alpha}{2\alpha+d}+\frac{d}{\alpha}} \text{ and } n \gtrsim \frac{1}{\epsilon^2}\left(\log(1/\delta) + \log\log n\right).
\tag{18}
$$

Next, we derive an explicit formula of the sample complexity satisfying Equation (15). Using Equations (14), (18), and (16), we have that $n$ satisfies Equation (15) if

$$
\begin{cases}
n &\gtrsim \frac{1}{\epsilon^2}\left[n^{\frac{2\beta+1}{2}\frac{d}{2\alpha+d}}(\log^2 n + \log(1/\epsilon))\right], \\
n &\gtrsim \left(\frac{1}{\epsilon^2}\right)^{1+\frac{d}{\alpha}}, \\
n &\gtrsim \frac{1}{\epsilon^2}\left(\log(1/\delta) + \log\log n\right).
\end{cases}
\tag{19}
$$

Note that $\beta \leq 1/2$ and $\frac{d}{\alpha} \leq 2$; thus, we have

$$\left(1 - \frac{2\beta + 1}{2} \frac{d}{2\alpha + d}\right)^{-1} \leq 1 + \frac{d}{\alpha} \leq 3.$$

Hence, $n$ satisfies Equations (18) and (19) if

$$n \gtrsim \left(\frac{1}{\epsilon^2}\right)^{1 + \frac{d}{\alpha}} \log^6 n + \frac{1}{\epsilon^2}(\log(1/\delta) + \log \log n).$$

## B    Technical Lemmas

**Lemma B.1** (Bartlett et al. (2005))**.** *Let $r > 0$ and let*

$$\mathcal{F} \subseteq \{f : \mathcal{X} \to [a, b] : \mathbb{V}[f(X_1)] \leq r\}.$$

*1. For any $\lambda > 0$, we have with probability at least $1 - e^{-\lambda}$,*

$$\sup_{f \in \mathcal{F}} (\mathbb{E}f - \mathbb{E}_n f) \leq \inf_{\alpha > 0} \left(2(1 + \alpha)\mathbb{E}\left[R_n \mathcal{F}\right] + \sqrt{\frac{2r\lambda}{n}} + (b - a)\left(\frac{1}{3} + \frac{1}{\alpha}\right)\frac{\lambda}{n}\right).$$

*2. With probability at least $1 - 2e^{-\lambda}$,*

$$\sup_{f \in \mathcal{F}} (\mathbb{E}f - \mathbb{E}_n f) \leq \inf_{\alpha \in (0,1)} \left(\frac{2(1 + \alpha)}{(1 - \alpha)}\mathbb{E}_\sigma\left[R_n \mathcal{F}\right] + \sqrt{\frac{2r\lambda}{n}} + (b - a)\left(\frac{1}{3} + \frac{1}{\alpha} + \frac{1 + \alpha}{2\alpha(1 - \alpha)}\right)\frac{\lambda}{n}\right).$$

*Moreover, the same results hold for $\sup_{f \in \mathcal{F}} (\mathbb{E}_n f - \mathbb{E}f)$.*

**Lemma B.2** (Györfi et al. (2002, Theorem 11.6))**.** *Let $B \geq 1$ and $\mathcal{F}$ be a set of functions $f : \mathbb{R}^d \to [0, B]$. Let $Z_1, ..., Z_n$ be i.i.d. $\mathbb{R}^d$-valued random variables. For any $\alpha > 0$, $0 < \epsilon < 1$, and $n \geq 1$, we have*

$$P\left\{\sup_{f \in \mathcal{F}} \frac{\frac{1}{n}\sum_{i=1}^n f(Z_i) - \mathbb{E}[f(Z)]}{\alpha + \frac{1}{n}\sum_{i=1}^n f(Z_i) + \mathbb{E}[f(Z)]} > \epsilon\right\} \leq 4\mathbb{E}N(\frac{\alpha\epsilon}{5}, \mathcal{F}|Z_1^n, n^{-1}\|\cdot\|_1)\exp\left(\frac{-3\epsilon^2\alpha n}{40B}\right).$$

**Lemma B.3** (*Contraction property* (Rebeschini, 2019))**.** *Let $\phi : \mathbb{R} \to \mathbb{R}$ be a L-Lipschitz, then*

$$\mathbb{E}_\sigma R_n \left(\phi \circ \mathcal{F}\right) \leq L\mathbb{E}_\sigma R_n \mathcal{F}.$$

**Lemma B.4** (Lei et al. (2016, Lemma 1))**.** *Let $\mathcal{F}$ be a function class and $P_n$ be the empirical measure supported on $X_1, ..., X_n \sim \mu$, then for any $r > 0$ (which can be stochastic w.r.t $X_i$), we have*

$$\mathbb{E}_\sigma R_n \{f \in \mathcal{F} : \|f\|_n^2 \leq r\} \leq \inf_{\epsilon > 0} \left[\mathbb{E}_\sigma R_n \{f \in \mathcal{F} - \mathcal{F} : \|f\|_\mu \leq \epsilon\} + \sqrt{\frac{2r \log N(\epsilon/2, \mathcal{F}, \|\cdot\|_n)}{n}}\right].$$

**Lemma B.5** (Lei et al. (2016, modification))**.** *Let $X_1, ..., X_n$ be a sequence of samples and $P_n$ be the associated empirical measure. For any function class $\mathcal{F}$ and any monotone sequence $\{\xi_k\}_{k=0}^\infty$ decreasing to 0, we have the following inequality for any non-negative integer $N$*

$$\mathbb{E}_\sigma R_n \{f \in \mathcal{F} : \|f\|_n \leq \xi_0\} \leq 4\sum_{k=1}^N \xi_{k-1} \sqrt{\frac{\log \mathcal{N}(\xi_k, \mathcal{F}, \|\cdot\|_n)}{n}} + \xi_N.$$

**Lemma B.6** (*Pollard's inequality*)**.** *Let $\mathcal{F}$ be a set of measurable functions $f : \mathcal{X} \to [0, K]$ and let $\epsilon > 0, N$ arbitrary. If $\{X_i\}_{i=1}^N$ is an i.i.d. sequence of random variables taking values in $\mathcal{X}$, then*

$$P\left(\sup_{f \in \mathcal{F}} \left| \frac{1}{N} \sum_{i=1}^N f(X_i) - \mathbb{E}[f(X_1)] \right| > \epsilon \right) \leq 8\mathbb{E}\left[N(\epsilon/8, \mathcal{F}|_{X_{1:N}})\right] e^{\frac{-N\epsilon^2}{128K^2}}.$$

**Lemma B.7** (*Properties of (bracketing) entropic numbers*)**.** *Let $\epsilon \in (0, \infty)$. We have*

1. $H(\epsilon, \mathcal{F}, \|\cdot\|) \leq H_{[]}(2\epsilon, \mathcal{F}, \|\cdot\|)$;

2. $H(\epsilon, \mathcal{F}|\{x_i\}_{i=1}^n, n^{-1/p} \cdot \|\cdot\|_p) = H(\epsilon, \mathcal{F}, \|\cdot\|_{p,n}) \leq H(\epsilon, \mathcal{F}|\{x_i\}_{i=1}^n, \|\cdot\|_\infty) \leq H(\epsilon, \mathcal{F}, \|\cdot\|_\infty)$ *for all* $\{x_i\}_{i=1}^n \subset dom(\mathcal{F})$.

3. $H(\epsilon, \mathcal{F} - \mathcal{F}, \|\cdot\|) \leq 2H(\epsilon/2, \mathcal{F}, \|\cdot\|))$, *where* $\mathcal{F} - \mathcal{F} := \{f - g : f, g \in \mathcal{F}\}$.

**Lemma B.8** (*Entropic number of bounded Besov spaces* (Nickl & Pötscher, 2007, Corollary 2.2))**.** *For $1 \leq p, q \leq \infty$ and $\alpha > d/p$, we have*

$$H_{[]}(\epsilon, \bar{B}_{p,q}^\alpha(\mathcal{X}), \|\cdot\|_\infty) \lesssim \epsilon^{-d/\alpha}.$$

**Lemma B.9** (*Approximation power of deep ReLU networks for Besov spaces* (Suzuki, 2018, a modified version))**.** *Let $1 \leq p, q \leq \infty$ and $\alpha \in (\frac{d}{p \wedge 2}, \infty)$. For sufficiently large $N \in \mathbb{N}$, there exists a neural network architecture $\Phi(L, m, S, B)$ with*

$$L \asymp \log N, m \asymp N \log N, S \asymp N, \text{ and } B \asymp N^{d^{-1} + \nu^{-1}},$$

*where $\nu := \frac{\alpha - \delta}{2\delta}$ and $\delta := d(p^{-1} - (1 + \lfloor \alpha \rfloor)^{-1})_+$ such that*

$$\sup_{f_* \in \bar{B}_{p,q}^\alpha(\mathcal{X})} \inf_{f \in \Phi(L, W, S, B)} \|f - f_*\|_\infty \lesssim N^{-\alpha/d}.$$

