# OpenReview forum: "On Sample Complexity of Offline Reinforcement Learning with Deep ReLU Networks in Besov Spaces"
_TMLR — Accepted by TMLR_

### Review · Reviewer_saLt · 2022-09-30

**Summary Of Contributions:**

The authors proposed to perform offline reinforcement learning in Besov space, which can be represented with deep ReLU networks under certain sparsity and norm constraints. Under the assumption that the function class is Bellman complete, the authors show the PAC guarantee of the FQI algorithm for both offline policy evaluation and offline policy learning. The main technical novelty is a uniform concentration on the $Q$ estimator for each iteration, which avoids the data-splitting.

**Broader Impact Concerns:**

This work focus on the theoretical analysis, so I believe it does not require any broader impact statement.

**Requested Changes:**

* If the authors claim the data-splitting will lead to severe issues, the authors should explicitly claim the dependency on $K$. If I understand correctly, we only require $K = O(\log(1/\varepsilon))$ to make the algorithmic error at a order of $\varepsilon$, which, for me is not an issue in both theory and practice.
* For the sub-optimality of offline learning, a proper definition should be $V^{\hat{\pi}}(s)$. I don’t see the reason we should use $Q^*$, as in the following steps we don’t follow $\pi^*$ as well.
* In the definition of the moduli of continuity and the Besov space, you already assume $\mathcal{X} = \mathbb{R}$, as the difference you use is $0\leq h \leq t$ instead of any other increment. I would like to argue that the authors need to discuss the topology of $\mathcal{X}$ at the beginning. This makes me puzzled as in reinforcement learning our state and action are never one dimensional.
* The statements in Remark 3.1 are repeating.
* The claim on the hardness of the correlated data claimed in Section 4 is hard to follow. I guess the authors would like to argue that $Q$ is estimated with $x_i$ in the last iteration, but we can decouple this dependency via a uniform convergence on $Q$.
* I don't understand the reason we don't return $V^\pi$ with the Bellman equation in Line 7 of the Algorithm 1.
* I’m not aware of any provable algorithms within polynomial time for optimizing sparse neural networks (with $\ell_0$ constraints), hence the claim in Section 4 is not proper. I hope the authors can provide references on that.
* Assumption 5.2 is just the Bellman completeness assumption that is widely used in reinforcement learning with general function approximation. I would claim that this assumption is in fact hiding lots of the issues, and make the original problem back to a statistical estimation problem with Besov space.
* Example 5.3 is not an example.




**Strengths And Weaknesses:**

Strengths:
* Extend the scheme of function approximation from reproducing kernel Hilbert spaces and Hölder spaces to Besov spaces.
* Provide a new method to decouple the dependency of $Q$ estimate with the data.

Weaknesses:
* The technical part is not so novel. It basically integrates the technical components of the statistical analysis on Besov spaces to standard offline reinforcement learning with general function approximation. The main technical improvement for reinforcement learning is on decoupling the $Q$ estimate from the data, which is relatively straightforward with the idea of uniform convergence.
* The presentation is poor.

---

> ### Author Response · Authors · 2022-10-07
> **Thank you for your constructive feedbacks and suggestions. Please see our response below**
>
> > If the authors claim the data-splitting will lead to severe issues ...  in both theory and practice.
>
> We remark that making the algorithmic error at a order of $\epsilon$ requires $K = \Omega(H \log(1/\epsilon))$ where $H = (1-\gamma)^{-1}$ is the effective horizon length. We revised our paper to include explicit $K$ and parameter $H$ in the comparison and discussion. In short, with such dependence inclusion, we conclude that data reuse in Algorithm 1  has a strong advantage over data splitting in Yang et al 2019 in long (effective) horizon problems where $H = \Omega(\log(1/\epsilon))$.
>
> ---
>
> > For the sub-optimality of offline learning ... we don’t follow $\pi^*$ as well.
>
> We used $Q^*$ by adopting the conventional metric of FQI-based estimators from Le et al. (2019), Yang et al. (2019), Munos & Szepesvari (2008). In fact, all the bounds in our paper are satisfied for $V^{\hat{\pi}}$ and we revised our paper to reflect your suggestion.
>
> ---
>
> > In the definition of the moduli of continuity ... as in reinforcement learning our state and action are never one dimensional.
>
> We used the definition of *univariate* moduli of smoothness in our initial version. We have revised to *multivariate* moduli of smoothness in our updated version.
>
> ---
>
> > The statements in Remark 3.1 are repeating.
>
> We have fixed it. Thanks!
>
> ---
>
> > The claim on the hardness of ... via a uniform convergence on $Q$
>
> We have clarified this in our updated version where the data dependence is due to that $Q_{k-1}$ also depends on $(s_i, a_i)$ making $\mathbb{E} [r_i + \gamma \max_{a} Q_{k-1}(s'_i, a)  ] \neq [T^*Q_{k-1}](s_i, a_i)$.
>
> We decoupled this dependence via a "double" uniform convergence argument over both $Q_{k-1}$ and $T^* Q_{k-1}$ when we bound an empirical process using local Rademacher complexities. This "double" uniform convergence argument is particularly helpful in dealing with local Rademacher complexities as local Rademacher complexities already involve the supremum operator which can be naturally incorporated with the "double" uniform convergence argument.
>
> ---
>
> > I don't understand ... in Line 7 of the Algorithm 1
>
> Line 7 presents OPE (Off-Policy Evaluation) where we are interested in estimating the value function $V^{\pi} = \mathbb{E}_{s \sim \rho, a \sim \pi(\cdot|s)} \left[Q^{\pi}(s,a) \right]$. We use $Q_K$ to esimate $Q^{\pi}$, thus use $V_K$ to estimate $V^{\pi}$. Note that we use $V^\pi$, not $V^k$, to compute the Bellman target for OPE in Line 3. We have revised the algorithm presentation to make it clearer in our updated version.
>
> ---
>
> > I’m not aware of ... can provide references on that.
>
> We indeed find no work that shows provable guarantees for optimizing $l_0$-constrainted neural networks and this work (Christos Louizos, Max Welling, and Diederik P. Kingma. Learning sparse neural networks through l0 regularization, 2017) provides a practical algorithm without guarantees. In practice, we might want to relax such $l_0$ constraint into a smooth objective and (S)GD can provably converge to the global optimum again. We have corrected our discussion regarding this part in our updated version.
>
> ---
>
> > Assumption 5.2 is just ... Besov space.
>
> We would like to remark that the completeness-type assumption is a common assumption and we argue for its necessity for MDP with function approximation as [Ruosong Wang, Dean P Foster, and Sham M Kakade. What are the statistical limits of offline rl with linear function approximation?] show the statistical hardness of offline RL with linear representation suggesting that only realizability and strong uniform data coverage are not sufficient for sample-efficient offline RL. We would like to make two remarks. The first one is that even though Assumption 5.2 imposes a regularity constraint into the underlying MDP, as we discussed in our paper, even the sufficient condition for Assumption 5.2 (that the reward function and transition density functions belong to a Besov space) already implies a large class of MDPs since Besov regularity is very general that to the best our knowledge we are not aware of any other function class used in the RL literature that is more general than Besov space. Second, though the completeness-type assumption helps leverage a statistical estimation problem, there are still unique challenges for esitmation in RL, notably the problem of error propagation over a long (effective) horizon. Our result suggests that our learning bound does not only hold for more general cases but also has a strong advantage in long (effective) horizon problems where the future returns are well delayed over time.
>
> ---
>
> > Example 5.3 is not an example.
>
> We have fixed that in our updated paper. Thanks!

---

### Review · Reviewer_2FTk · 2022-10-02

**Summary Of Contributions:**

This paper studies the sample complexity of offline policy evaluation/optimization under deep ReLu neural network approximation. The theoretical analysis is conducted under two key assumptions: (1) "Besov dynamic closure" and (2) behavior policy distribution coverage and the proposed algorithm adopts a standard LSQI structure. Under those two assumptions, the author established the suboptimality guarantee of their proposed algrithm in OPL and OPE problems. The author also compare their theoretical results with some state-of-the-art results that under tabular or linear MDP assumptions to demonstrate the significance of their work.

**Broader Impact Concerns:**

This is a theoretical work and does not hav significant ethical concern.

**Requested Changes:**

See my comments in weakness.

**Strengths And Weaknesses:**

Strengths:

(1) Offline RL with nonlinear function approximation is a very challenging problem. Under practical settings, neither the convergence nor the global optimality guarantee can be guaranteed. This paper provides a new framework to establish some interesting results.
(2) Overall, this paper is easy to follow.
(3) The theoretical result guarantee and algorithm structure have some advantage over some previous works.

Weakness:
(1) Although assumption 2 is acceptable given the challenging setting this paper cares about, assumption 1 is a little bit strong compare with many SOTA works, since they only requie nonzero distribution coverage in state-action pairs that are also non-zero in target/optimal policy distributions. Given assumption 2 in this paper, many technical difficulties can be avoided. Although it might be the case that under the deep nerual network and Besove assumptions previous coverage assumption may not be able to lead to a reasonably good bound. If that is the case, the author might need to add some comments here to illustrate why previous coverage assumption can not be adopted here.

(2) The "uniform bound" techniques have been widely used in previos regret based RL analysis to handle the dependence between samples. The "double uniform convergence argument" in this paper is more like a nature adaptation of previous work due to the nature of the problem instead of proposing a new technique. The author might need to further demonstrate their novel regarding this.

(3) When comparing with other works, it might be fair to include one column of coverage assumption that they adopted.

---

> ### Author Response · Authors · 2022-10-07
> **Thank you for your constructive feedbacks and suggestions. Please see our response below**
>
> > Weakness (1)
>
> We remark that there are two main approaches to OPE/OPI with function approximation, either uncertainty-agnostic and uncertainty-aware methods. The uncertainty-agonistic methods such as FQI-type estimators directly learn state-action value functions from the offline data without constructing any statistical confidence regions (either explicitly or implicitly). This family of methods requires the offline data to have a uniform coverage over the state-action space as Assumption 1. The uncertainty-aware methods such as the recent line of works in pessimism for offline RL [Jin et al. (2020b)] only need the offline data to cover the trajectory of the target or optimal policy. That said, we can include the pessimistic approach to the current work with a more involved analysis of uncertainty quantifiers under non-linear function approximation to reduce the strictness of the data coverage assumption. We leave this extension to future work and focus instead on the simplest setting of offline RL to understand the fundamental issue of sample complexity of offline RL with neural network function approximation under a general regularity defined by Besov spaces. We have reflected this discussion in our updated version.
>
> ---
>
> > Weakness (2)
>
> We remark that the double uniform convergence argument is a natural extension of the uniform convergence argument in standard statistical learning theory (for regression) in hindsight. This double uniform convergence argument is particularly helpful in dealing with local Rademacher complexities under a data-dependent structure as local Rademacher complexities already involve the supremum operator which can be naturally incorporated with the double uniform convergence argument. We believe this idea could be used in broader contexts. Besides, the adoption of local Rademacher complexities and a uniform convergence variant of Bernstein’s inequality for analyzing offline RL with function approximation under a regularity of the Bellman operator are also later adopted by other recent works. We have revised our paper to make it clearer our proof approach and replaced "novelty" with softer language in our updated version
>
> ---
>
> > Weakness (3)
>
> In our paper and Table 1, we only compare with works that rely on uniform data coverage and we leave the incorporation of pessimism into our framework for partial data coverage to a future work.

---

### Review · Reviewer_HrkQ · 2022-10-03

**Summary Of Contributions:**

This paper studies the problem of sample complexity for offline RL (OPE and OPL) using neural network function approximators. In particular, the authors study the general case of neural networks satisfying a Besov condition, which is more general than prior analyses. Sample complexity bounds are presented for OPE and OPL and they are comparable to existing works. The underlying Besov condition, which replaces a typical completeness condition, requires that for any $f \in \mathcal F$, $T^\pi f$ is in the Besov space. The analysis avoids sample splitting.

**Broader Impact Concerns:**

I do not have any concerns.

**Requested Changes:**

_Major things_

- There is one important question which I feel is not properly addressed in the main text. This paper claims that the condition subsumes all previously made completeness conditions and goes on to state model-based assumptions like the linear MDP. But such model-based assumptions are generally not the ones used to analyze offline RL problems. Existing literature is often concerned with more generic completeness conditions like $f \in \mathcal F \implies T^* f \in \mathcal F$, which doesn’t explicitly constrain the MDP (only implicitly). It has been shown many times that this along with Assumption 5.1 is sufficient (Chen and Jiang, 2019) and more recently it has been shown that sample efficient learning is not possible without it (in general). If completeness is not fully satisfied, one usually sees some approximation error term show up. So my question is the following. Does Assumption 5.2 and the requirement in the second part of Theorem 5.1 implicitly constrain the problem so that the approximation error is zero? Judging from the analysis, it seems this is indeed the case. It also seems to be the case in Yang et al. I hope the authors can clarify. If the answer is yes, I believe it is necessary to be upfront about this since these subtle parts of the assumptions are of great importance in the surrounding literature.

- Additionally this seems to lead to a chicken-and-egg problem: Assumption 5.2 is assumed to hold at the start of Theorem 5.1 (presumably with $\alpha$, $p$, and $q$ set according to the defined class $\mathcal F$). But at the end of the theorem statement, the parameters of $\mathcal F$ are chosen according to $\alpha$. So my question to the authors is: how should we interpret the “order of operations” here. What should really go first? Should we assume that all functions $f \in (\mathcal S  \times \mathcal A \to \mathbb R)$ satisfy $T^\pi f \in B$ for this particular MDP but it is sufficient to consider the restriction to $\mathcal F$ with those parameters in Theorem 5.1? Or should we say that $\mathcal F$ is given and happens to satisfy 5.2 and the conditions in Theorem 5.1 simultaneously?

The latter seems more reasonable to me. But then I think the statement of Assumption 5.2 should be corrected. It should perhaps say there exists p, q, \alpha such that (L, m, S, B) satisfy those conditions originally stated in Theorem 5.1 and T^\pi \in B. This is less general than before, but, in my view, this is the correct way.

- The exponential dependence on d is a bit jarring but there is also a requirement that $\alpha = \Omega (d)$ in Assumption 5.2, which somehow makes this okay. But I am wondering if this requirement is just trivializing problem. Can the authors clarify this part?

_Minor things_

- Typical completeness conditions require that $f \in \mathcal F \implies T^* f \in \mathcal F$ but Assumption 5.2 says that this must hold for all $T^\pi$ operators. Is this really necessary? Why?

- The claims about K -> inf in Yang et al are a bit strong. While this is undesirable practically, the language is a bit harsh. K need only be logarithmically large, which is really not that bad and you often see similar data splitting methods in many fields like robust estimation, so it is not totally unimaginable that someone would do this.

- “This correlated structure hinders a direct use of the standard concentration inequalities. We overcome this technical difficulty using uniform convergence argument.”

I do not think these two are mutually exclusive. You can use concentration inequalities to prove uniform convergence results…

- I’m a little skeptical about the claims of how novel this “double uniform convergence” analysis is. Prior works like Chen and Jiang (and earlier works referenced in their paper) have handled this and they did not give it special treatment. Granted, analyses are typically a lot less messy when the function classes are finite, but the principle is the same and I think this should be made clear in the paper.


**Strengths And Weaknesses:**

Strengths
- The paper proves a more general sample complexity result when $T^\pi f$ belongs to a Besov class whenever $f \in \mathcal F$. Specifically, prior works have shown similar results for Holder classes or with finite classes.
- There are comparatively fewer theoretical works addressing neural networks for offline RL despite widespread usage of them in practice and so it is timely and contributes to filling the knowledge gap.
- The rates seem to be very close to existing work, but with potentially more general conditions.
- The algorithm does not require sample splitting unlike prior work.

Weaknesses
- There are several notable clarity issues regarding the assumptions about completeness (5.2) and other claims. See questions in the next section.
- The analysis does not seem too novel (see also questions about this), but this isn’t a big deal as long as the claims are accurate, as per TMLR guidelines.
- It could be argued that 5.1 is strong, but, given that papers have only recently been addressing weaker coverage conditions and there are still many interesting problems involving 5.1, I think it is acceptable for now.

---

> ### Author Response · Authors · 2022-10-07
> **Thank you for your constructive feedbacks and suggestions. Please see our response below**
>
> > There is one important question ... in the surrounding literature
>
> We remark that our completeness-type assumption, unlike the completeness assumption $f \in \mathcal{F} \implies T^*f \in \mathcal{F}$, does not make the approximation error zero. Our assumption does not require closedness under $\mathcal{F}$ but only requires that the resulting function from the Bellman operator applied on a neural network is captured on the Besov space. Even though the assumption looks stringent at first as it requires over all $f \in \mathcal{F}_{NN}$, as we pointed out a sufficient condition for our assumption, if the transition kernel and the reward functions are Besov, the assumption is satisfied. Since the Besov space is very general that to the best of our knowledge, is more general than any function classes currently used in RL, even this sufficient condition is already mild enough in practice.
>
> ---
>
> > Additionally this seems to lead to a chicken-and-egg problem ...
>
> Thank you for pointing out this ambiguity. We confirm that the latter is the case. For Theorem 5.1. to hold, (along with assumption 5.1.) we only need Assumption 5.2 holds for a particular $\mathcal{F}$ with its parameters $L,m,S,B$ specified in Theorem 5.1. Moreover,  the sufficient condition for Assumption 5.2 is that when the transition kernel and reward functions are on the Besov space, regardless of any function approximator $\mathcal{F}$ or policy $\pi$ we use in Assumption 5.2, Assumption 5.2 holds. This suggests that we can use the aforementioned sufficient condition to replace the initial Assumption 5.2 to remove the dependence on any network architecture. To maintain the generality in our result, we have decided not to do so. We have clarified this in our updated version per your suggestion.
>
> ---
>
> > The exponential dependence on d is a bit jarring  ...
>
> The requirement $\alpha > d/2$ is necessary to guarantee compactness (i.e. its log covering number is finite) of the Besov space. To the best of our knowledge, we are not aware of any condition other than the stated one to guarantee the compactness of the Besov space.
>
> ---
>
> > Typical completeness conditions require that ... Is this really necessary? Why?
>
> This condition is in fact stronger than what we need in our proof. We only need it to hold for $T^*$ in OPL task and $T^{\pi}$ for OPE task where $\pi$ is the target policy to be evaluated.  We have revised this part in our updated version.
>
> ---
>
> > The claims about K -> inf in Yang et al are a bit strong ...
>
> We agree this is quite a harsh comparison when letting K -> inf.  As with our response to the similar question raised by Reviewer saLt, we remark that we need $K = {\Omega}(H \log(1/\epsilon))$ where $H = (1-\gamma)^{-1}$ is the effective horizon length to make algorithmic error at an order of $\epsilon$. We revised our paper to include explicit $K$ and parameter $H$ in the comparison and discussion. In short, with such dependence inclusion, we conclude that data reuse in Algorithm 1 has a strong advantage over data splitting in Yang et al 2019 in long (effective) horizon problems where $H = \Omega(\log(1/\epsilon))$.
>
> ---
>
> > “This correlated structure hinders a direct use of the standard concentration inequalities. We overcome this technical difficulty using uniform convergence argument.”
>
> As the standard concentrations such as Bernstein's inequality require the random variables to adapt to some filtration. The correlated structure breaks the condition of adaption to filtration. We have revised this sentence in our updated version to remove its confusion per your suggestion.
>
> ---
>
> > I’m a little skeptical about the claims  ...
>
> We were not aware of the “double” idea analysis in Chen and Jiang paper at the time when we initially wrote our paper. But we would like to remark that while the general idea of “double uniform convergence” is natural in hindsight, this is particularly helpful in our analysis when we bound the local Rademacher complexity of complex function approximations and data-dependent structure  as local Rademacher complexities already involve the supremum operator which can be naturally incorporated with the double uniform convergence argument.
> We believe this idea could be used in broader contexts. We have revised our paper to make it clearer our proof approach, replaced "novelty" with softer language , and discussed Chen and Jiang paper in our updated version.

---

### Author Response · Authors · 2022-10-07
**Thank you so much for your constructive feedbacks. We have responded to all the questions and revised our paper accordingly**

Dear our Reviewers and Editors,

Thank you so much for taking the time to handle and review our paper and providing constructive feedbacks. We have responded to all questions in the reviews individually. To facilitate any of our further discussion, we briefly summarize the key questions as follows:

- Comparison with data splitting in Yang et al with explicit $K$
- Clarification of "double" uniform convergence argument
- Clarification of the completeness-type Assumption 5.2 (along with the specification of network architecture in the assumption)
- Clarification of Assumption 5.1 and position with recent pessimistic algorithms

Please find our responses to these questions in our individual replies below.

We have also revised our paper to reflect our reviewers' suggestions in our updated version. The revised parts are highlighted in blue color. Please let us know if you have further questions regarding our responses and our work.


Thank you.

Best,
Authors

---

### Decision · Action_Editors · 2022-12-07

**Recommendation:** Accept with minor revision

**Comment:**

The reviewers have conflicting views about the novelty and the impact of the work, but all agree that the paper is a solid contribution. The primary concerns are threefold.  1. The completeness assumption is strong (stronger than existing work that operates in simpler settings).  2. The "double uniform convergence" technique is not as novel as the authors indicated initially.  3.  The exponential dependence on the dimension is unsatisfactory.

I find the first two concerns valid, but the authors have submitted a revision that clearly carves out a niche for them and had a more modest claim on the novelty. I think the revision is satisfactory.  There is nothing the authors could do for the third concern because Besov space is so large that the "curse of dimension" is unavoidable even in simpler supervised learning settings.  In some sense, only because Besov space is so large, it is possible for functions --- after Bellman backups --- could possibly be still in the space.

Under these arguments, I think that the paper passes the correctness and relevance bar for TMLR and the results should be made known to RL theory people.  I think the authors could have done a better job in giving examples of Besov space functions so as to motivate the study a bit more. For example, whenever "max" is involved, it is hard to argue that the resulting function out of the Bellman operator is still within the same smooth function class, because it essentially becomes piecewise. Yet, it could still be within the Besov function with a small \alpha parameter.

I have one more concern about the specific assumptions and one other item on my wishlist that I hope the authors discuss in the final version of the paper.

1. In Assumption 5.2, the claim is that for any neural network, after passing it through a Bellman operator, it sits inside a Besov class.  However, even neural networks are not in Besov classes with \alpha > 1 (which requires higher order differentiability).   NN can "improperly" approximate Besov class functions of any order of smoothness in L_infty but cannot exactly recover Besov space functions.

2. The authors should have discussed the optimality of the rates.  It is hard for me to read it off from the current results. It appears to be suboptimal?  If the authors can construct a lower bound showing that the current rate is minimax optimal then the results would be significantly stronger.  It is not required because the current paper passes the bar, but I think it is a result that should naturally go into the same paper if easily provable.


**Audience:**

The paper is of interest to many readers of TMLR. The primary audience is researchers who work on RL theory, and statisticians who are interested in getting into RL.

**Claims And Evidence:**

The paper studies offline reinforcement learning with nonparametric function approximation. Specifically, the value function of any policies is assumed to be in a Besov space and by the results of Suzuki (2018) a sufficiently wide and deep feedforward NN with a particular sparsity level can be used to approximate and estimate functions in a Besov space.

The main results of this paper is the statistical rates, i.e., sample complexity of offline RL under this new setting.  Existing work focused on smaller and more restrictive function classes such as Holder class and RKHS.

The reviewers, who checked the proofs, found the results to be correct.  I have some additional comments that I hope the authors could address in the final version (see the "comment" block below).

---

> ### Author Response · Authors · 2022-12-11
> **Thank you for the decision and additional comments.**
>
> Dear Editor and our reviewers,
>
> Thank you for your time in steering the discussion and making the decision on our paper.  We're excited.
>
> > Concern about Assumption 3.1
>
> We appreciate your raising this concern. We were also aware of the need to clearly explain the rationale and motivation behind why Assumption 3.1 holds in the initial phase of this project and several rounds of discussion have helped us consolidate it further. One way to explain why Assumption 3.1 holds is by using our sufficient condition of Assumption 3.1, which is stronger yet much more intuitive. Specifically, the sufficient condition is that the expected reward function $r(\cdot)$ and the density function of the transition kernel $g_{s'}(\cdot) := P(s'| \cdot )$ both sit in the Besov space, for any $s' \in \mathcal{S}$, as the Bellman operator is linear in both $r(\cdot)$ and $g_{s'}(\cdot)$. This is similar to the way linear MDP imposes linearity on $r(\cdot)$ and $g_{s'}(\cdot)$ as, $r(s,a) = \phi(s,a)^T w$ and $g_{s'}(\cdot) = \phi(s,a)^T \nu(s') $, where $\nu(s') = (\nu_1(s'), \ldots, \nu_d(s'))$ and $\nu_i(s')$ is some signed measure over $\mathcal{S}$. Our sufficient condition above imposes the smoothness constraint solely on the underlying MDP regardless of the input function f. Thus, the “max” over the input function f(s, a) does not affect the smoothness of the resulting function after f is passed
> through the Bellman operator.  With that sufficient condition, Assumption 3.1 holds regardless of whether f is in the Besov space.
>
> > The optimality of the rates
>
> Though our bound has a tighter dependence on the effective horizon H in the long horizon setting, the dependence on ϵ in our bound is compromised due to data reuse and thus does not match the minimax rate in the (supervised) regression setting.  We leave as future direction to construct the lower
> bound for the data-reuse setting of non-parametric offline RL, which, to the best of our knowledge, we are not aware of any such result in the literature.
>
> We have incorporated these points in the final version of our paper. We're happy to address any further concerns.
>
> Best,
> Authors.